# Long noncoding RNA Malat1 protects against osteoporosis and bone metastasis

Yang Zhao [1,10], Jingyuan Ning[2,10], Hongqi Teng[1], Yalan Deng [1], Marisela Sheldon[1], Lei Shi[3], Consuelo Martinez[1], Jie Zhang[1], Annie Tian[4], Yutong Sun[5], Shinichi Nakagawa [6], Fan Yao [1,9], Hai Wang [7] & Li Ma [1,8] ✉

MALAT1, one of the few highly conserved nuclear long noncoding RNAs (lncRNAs), is abundantly expressed in normal tissues. Previously, targeted inactivation and genetic rescue experiments identified MALAT1 as a suppressor of breast cancer lung metastasis. On the other hand, Malat1-knockout mice are viable and develop normally. On a quest to discover the fundamental roles of MALAT1 in physiological and pathological processes, we find that this lncRNA is downregulated during osteoclastogenesis in humans and mice. Remarkably, Malat1 deficiency in mice promotes osteoporosis and bone metastasis of melanoma and mammary tumor cells, which can be rescued by genetic add-back of *Malat1*. Mechanistically, Malat1 binds to Tead3 protein, a macrophage-osteoclast–specific Tead family member, blocking Tead3 from binding and activating Nfatc1, a master regulator of osteoclastogenesis, which results in the inhibition of Nfatc1-mediated gene transcription and osteoclast differentiation. Notably, single-cell transcriptome analysis of clinical bone samples reveals that reduced MALAT1 expression in pre-osteoclasts and osteoclasts is associated with osteoporosis and metastatic bone lesions. Altogether, these findings identify Malat1 as a lncRNA that protects against osteoporosis and bone metastasis.

Osteoporosis, characterized by decreased bone mineral density (BMD), increased bone fragility, and susceptibility to fracture, reflects an imbalance in which osteoclastic bone resorption exceeds osteoblastic bone formation[1,2] and is a potential contributor to acceleration of bone metastasis[3–5]. Primary osteoporosis occurs during the aging process, particularly in postmenopausal women[6]. Secondary osteoporosis has the same outcome as primary osteoporosis but is caused by certain medical conditions or medications[7]. In either condition,

excessive osteoclastogenesis plays a key role and provides opportunities for therapeutic intervention.

Osteoclasts, a class of multinucleated giant cells (MGCs) that originate from the monocyte/macrophage lineage, are responsible for the resorption of bone matrix and minerals[8,9]. Osteoclastogenesis is initiated by macrophage colony-stimulating factor (M-CSF) and receptor activator of nuclear factor-κB (RANK) ligand (RANKL), which induce the expression of osteoclast markers, such as cathepsin K

[1]Department of Experimental Radiation Oncology, The University of Texas MD Anderson Cancer Center, Houston, TX 77030, USA. [2]Institute of Basic Medical Sciences, Peking Union Medical College, Chinese Academy of Medical Sciences, Beijing 100010, China. [3]Department of Endocrine Neoplasia and Hormonal Disorders, The University of Texas MD Anderson Cancer Center, Houston, TX 77030, USA. [4]Department of Kinesiology, Rice University, Houston, TX 77005, USA. [5]Department of Molecular and Cellular Oncology, The University of Texas MD Anderson Cancer Center, Houston, TX 77030, USA. [6]RNA Biology Laboratory, Faculty of Pharmaceutical Sciences, Hokkaido University, Sapporo 060-0812, Japan. [7]Department of Molecular and Cellular Biology, Roswell Park Comprehensive Cancer Center, Buffalo, NY 14263, USA. [8]The University of Texas MD Anderson Cancer Center UTHealth Houston Graduate School of Biomedical Sciences, Houston, TX 77030, USA. [9]Present address: Hubei Hongshan Laboratory, College of Biomedicine and Health, Huazhong Agricultural University, Wuhan, Hubei 430070, China. [10]These authors contributed equally: Yang Zhao, Jingyuan Ning. ✉e-mail: lma4@mdanderson.org

(CTSK) and acid phosphatase 5 (ACP5, also known as TRAP), followed by maturation of osteoclast precursors and cell-cell fusion[10]. As a master regulator of osteoclastogenesis, the nuclear factor of activated T cells 1 (NFATC1) is induced by RANKL, which in turn forms a complex with other transcription factors[11] and activates the transcription of its own coding gene as well as other genes involved in osteoclast adhesion, cell fusion, and bone resorption[12–14].

Long noncoding RNAs (lncRNAs), transcripts that are longer than 200 nucleotides and are not translated into proteins, function through binding to DNA, other RNA, and proteins[15,16]. LncRNAs usually have low evolutionary conservation. One of the few exceptions, MALAT1, is a highly conserved nuclear lncRNA that is abundantly expressed in many tissues[17]. MALAT1 has been shown to modulate alternative pre-mRNA splicing based on siRNA knockdown results from cultured cell lines[18]. In 2012, three groups reported that Malat1-knockout mice showed no obvious phenotypic differences compared with wild-type mice under physiological conditions, and loss of Malat1 in mice did not affect alternative pre-mRNA splicing[19–21]. On the other hand, recent animal studies suggested that Malat1 has important functions under pathological conditions. For instance, through targeted inactivation, restoration (genetic rescue), and overexpression of Malat1 in mouse models, we found that Malat1 suppresses breast cancer lung metastasis through binding and inactivation of the Tead family of transcription factors[22]. Moreover, Malat1-null mice exhibited enhanced antiviral responses, suggesting that Malat1 may suppress antiviral innate immunity[23]. In addition, when fed a high-fat diet, Apoe[−/−] mice transplanted with Apoe[−/−];Malat1[−/−] bone marrow showed higher atherosclerotic plaque burden in the aorta and increased hematopoietic progenitor cells and their progeny, suggesting that Malat1 may regulate hematopoietic cells[24].

Recent genome-wide association studies (GWAS) showed that single-nucleotide polymorphisms (SNPs) are associated with osteoporosis[1,25]. Interestingly, one such analysis identified an SNP (rs202070768) at the MALAT1 locus that was associated with low BMD[26]. However, functional evidence of MALAT1 alterations having a role in low BMD and osteoporosis is lacking. In the present study, by using genetically engineered mouse models, we identify Malat1 as a negative regulator of osteoporosis and bone metastasis. Mechanistically, Malat1 binds and sequesters Tead3, blocking Tead3 from interacting with and activating Nfatc1. Consequently, loss of Malat1 derepresses Tead3, which in turn promotes Nfatc1-mediated osteoclast differentiation. Notably, single-cell RNA-seq analysis demonstrates an association of reduced MALAT1 expression in the osteoclast lineage with osteoporosis and bone metastasis.

## Results

### MALAT1 is downregulated during osteoclast differentiation in humans and mice

Hematopoietic stem cells (HSCs) undergo self-renewal and differentiation in the bone marrow. During a hierarchical differentiation process, HSCs turn into multipotent progenitors (MPPs), which then differentiate into oligopotent progenitors, including common myeloid progenitors (CMPs) and common lymphoid progenitors (CLPs)[27]. Recent reports of Malat1 having a role in regulating hematopoietic cells under pathological conditions[23,24] prompted us to analyze MALAT1 expression during differentiation of HSCs by using publicly available high-throughput sequencing datasets. Interestingly, in both humans and mice, MALAT1 was expressed at higher levels in HSCs than in MPPs or CMPs (Supplementary Fig. 1a–d).

CMPs can differentiate into monocytes and macrophages, which are the precursors of osteoclasts[28]. We analyzed gene expression during the differentiation of human placental CD14[+] macrophages into MGCs[29] (Fig. 1a), in which osteoclasts are the major cell population[8]. Compared with CD14[+] macrophages, MGCs showed elevated expression of osteoclast markers, including NFATC1, CTSK, DCSTAMP,

ATP6V0D2, ATP6V0E2, and ATP6V0A1 (Fig. 1b, c). In contrast, MALAT1 was significantly downregulated in MGCs relative to CD14[+] macrophages (Fig. 1a–c). Consistent with the functions of osteoclasts, gene set enrichment analysis (GSEA) indicated that the gene sets enriched in MGCs compared with CD14[+] macrophages were related to collagen organization, extracellular structure remodeling, and skeletal development (Fig. 1d and Supplementary Data 1). To further validate the downregulation of Malat1 during the differentiation of macrophages into osteoclasts, we treated a mouse macrophage/pre-osteoclast cell line, RAW264.7, with soluble RANKL to induce osteoclast differentiation[30]. After this treatment, markers of osteoclasts, including Nfatc1, Ctsk, and Trap5, were upregulated in a time-dependent manner (Fig. 1e–g), whereas Malat1 expression levels were markedly decreased (Fig. 1h). Taken together, these results reveal MALAT1 as a lncRNA that is downregulated during osteoclastogenesis in humans and mice.

Lipopolysaccharides (LPS) and TNF-α are involved in pathological osteoclastogenesis[30–33]. Consistent with previous reports[31,32], we observed that LPS treatment alone was insufficient to initiate osteoclast differentiation; instead, when RAW264.7 cells were pretreated with RANKL, LPS treatment promoted osteoclastogenesis, as gauged by staining for tartrate-resistant acid phosphatase (TRAP), a widely used marker of osteoclasts (Fig. 1i, j), and upregulated the expression of Nfatc1 and Ctsk (Fig. 1k). TNF-α, on the other hand, can induce osteoclast differentiation in both RANKL-dependent and RANKL-independent manners[34,35]. Indeed, we found that treating RAW264.7 cells with TNF-α induced osteoclastogenesis and the expression of Nfatc1 and Ctsk, either with or without RANKL pretreatment (Fig. 1i, l, m). Notably, Malat1 was substantially downregulated during LPS- or TNF-α-induced osteoclast differentiation (Fig. 1k, m). These findings suggest that Malat1 may be involved in osteoclastogenesis in response to various stimuli.

### Genetic models reveal that Malat1 protects against osteoporosis and bone metastasis

To study the role of Malat1 in osteoclastogenesis and osteoporosis in vivo, we used a Malat1-knockout mouse model (Malat1[−/−]) described in our previous study, in which a transcriptional terminator was inserted downstream of the transcriptional start site of Malat1, causing the loss of Malat1 RNA expression without altering expression levels of Malat1's adjacent genes[22]. Also, we previously engineered mice with targeted transgenic Malat1 expression from the ROSA26 locus (Malat1[Tg/Tg]), which enabled us to conduct genetic rescue studies in Malat1[−/−] mice by generating Malat1[−/−];Malat1[Tg/Tg] animals[22]. We collected various tissues, including bone marrow, stomach, colon, small intestine, liver, and pancreatic tissues, from Malat1[+/+], Malat1[−/−], and Malat1[−/−];Malat1[Tg/Tg] mice and measured Malat1 expression levels by qPCR. This analysis confirmed Malat1 depletion in Malat1[−/−] mice and its re-expression in Malat1[−/−];Malat1[Tg/Tg] mice, although the levels of Malat1 restoration varied among different tissues (Supplementary Fig. 2a).

By performing microcomputed tomographic (μCT) analysis of the femurs of 6-month-old animals, we found that both male and female Malat1[−/−] mice had much lower bone density than age- and sex-matched Malat1[+/+] mice; importantly, this osteoporotic phenotype was rescued in Malat1[−/−];Malat1[Tg/Tg] mice (Fig. 2a, b). Quantification of the μCT data revealed that compared with Malat1[+/+] mice, the trabecular bone density (Fig. 2c, d), trabecular bone volume per tissue volume (BV/TV, Supplementary Fig. 2b, c), trabecular number (Tb.N, Supplementary Fig. 2d, e), and trabecular thickness (Tb.th, Supplementary Fig. 2f, g) were significantly reduced in Malat1[−/−] mice, and these reductions were largely reversed by genetic restoration of Malat1 expression (Fig. 2c, d and Supplementary Fig. 2b–g). Staining for TRAP revealed a significant increase in osteoclasts in the femurs of Malat1[−/−] mice compared with Malat1[+/+]

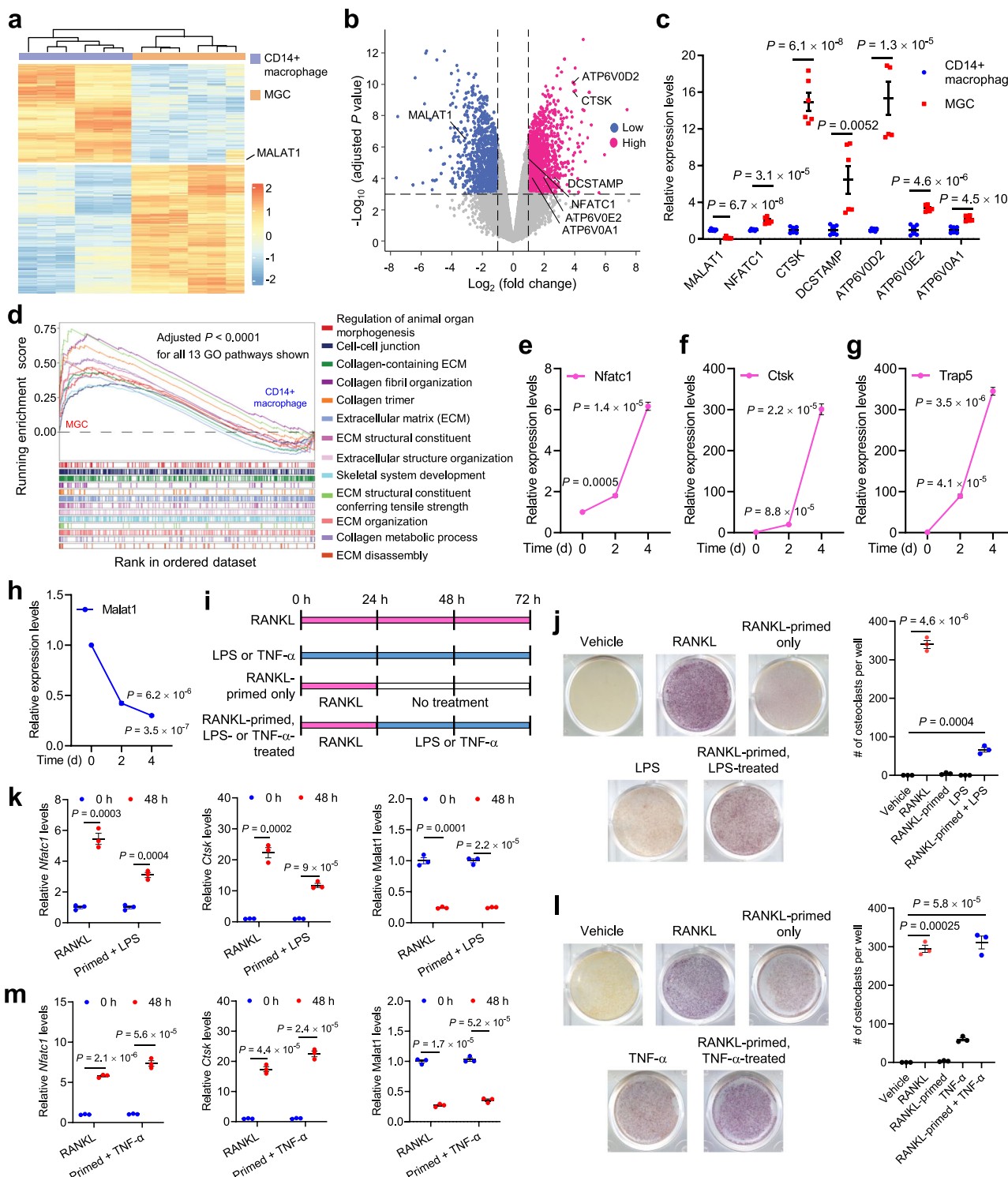

mice, and this increase was reversed in *Malat1⁻/⁻;Malat1*^Tg/Tg mice (Fig. 2e). Quantification of femoral osteoclasts showed that relative to *Malat1*^+/+ mice, the osteoclast numbers per bone perimeter (Oc.N/B.Pm, Fig. 2f) and the osteoclast surface per bone surface (Oc.S/BS, Fig. 2g) were elevated in *Malat1*⁻/⁻ mice by approximately 2-fold. Moreover, enzyme-linked immunosorbent assay (ELISA) revealed that *Malat1*⁻/⁻ mice had higher levels of the serum bone resorption marker TRAP5b than *Malat1*^+/+ and *Malat1*⁻/⁻;*Malat1*^Tg/Tg mice (Supplementary Fig. 2h).

Next, to determine the role of Malat1 in modulating pathological bone loss, we used a well-established mouse model of inflammatory bone resorption, which involves the injection of LPS into the subcutaneous space over the calvarial bones[36]. As gauged by μCT imaging, administration of LPS to 8-week-old *Malat1*⁻/⁻ mice resulted in significantly aggravated erosions on the surface of the calvarial bones, compared with *Malat1*^+/+ or *Malat1*⁻/⁻;*Malat1*^Tg/Tg mice (Fig. 2h, i). TRAP staining and quantification revealed that after LPS injection, *Malat1*⁻/⁻ mice had higher osteoclast numbers per bone perimeter (Oc.N/B.Pm, Fig. 2j, k) and more osteoclast surface per bone surface (Oc.S/BS, Fig. 2l) than either *Malat1*^+/+ or *Malat1*⁻/⁻;*Malat1*^Tg/Tg mice. Collectively, these findings indicate that Malat1 deficiency promotes osteoporosis under both physiological and inflammatory conditions.

**Fig. 1 | MALAT1 is downregulated during osteoclast differentiation. a–d** CD14[+] human placental macrophages were differentiated into multinucleated giant cells (MGCs) in culture. Both CD14[+] macrophages and MGCs were subjected to high-throughput RNA sequencing (RNA-seq). *n* = 6 biological replicates per group. Data source: GSE38747. **a** Heatmap of differentially expressed genes between CD14[+] macrophages and MGCs. **b** Volcano plot of genes upregulated (red) or down-regulated (blue) in MGCs relative to CD14[+] macrophages. Cutoff values: |log$_2$ (fold change)| >1 and adjusted *P* value < 0.001. Statistical significance was determined by a linear model with Benjamini–Hochberg correction. **c** Relative expression levels of MALAT1 and osteoclast markers were quantitated from the RNA-seq results. **d** Gene set enrichment analysis (GSEA) of the RNA-seq data, showing the top 13 Gene Ontology (GO) pathways. Statistical significance of the pathway enrichment score was determined by an empirical phenotype-based permutation test. Normalized enrichment scores (NES) and enriched pathways with an adjusted *P* value < 0.05 are listed in Supplementary Data 1. **e–h** qPCR of *Nfatc1* (**e**), *Ctsk* (**f**), *Trap5* (**g**), and Malat1 (**h**) in RAW264.7 cells treated with soluble RANKL (50 ng/mL) for the indicated times. *n* = 3 biological replicates per group. **i** Schematic representation of the treatments used to evaluate the osteoclastogenic activity of RANKL, LPS, and TNF-α, with or without pretreatment (priming) with RANKL. **j** TRAP staining images (left panel) and quantification (right panel) of RAW264.7 cells treated with RANKL or LPS, with or without pretreatment with RANKL. *n* = 3 wells per group. **k** qPCR of *Nfatc1*, *Ctsk*, and Malat1 in the cells described in **j**. *n* = 3 biological replicates per group. **l** TRAP staining images (left panel) and quantification (right panel) of RAW264.7 cells treated with RANKL or TNF-α, with or without pretreatment with RANKL. *n* = 3 wells per group. **m** qPCR of *Nfatc1*, *Ctsk*, and Malat1 in the cells described in **l**. *n* = 3 biological replicates per group. Statistical significance in **c**, **e–h**, and **j–m** was determined by a two-tailed unpaired *t* test. Error bars are s.e.m. Source data are provided as a Source Data file.

Untreated osteoporosis is associated with accelerated progression of bone metastasis in cancer patients[3–5]. Drugs for osteoporosis therapy, such as bisphosphonates that inhibit osteoclast-mediated bone resorption, have been used to treat bone diseases, including bone metastases[37]. To determine whether Malat1 in the host confers protection from bone metastases, we injected luciferase-labeled B16F1 melanoma cells into the tibiae of 6-month-old male *Malat1*[+/+], *Malat1*[−/−], or *Malat1*[−/−];*Malat1*[Tg/Tg] mice, and we found that bone metastases were markedly exacerbated by Malat1 loss in the hosts, a phenotype that was rescued by Malat1 re-expression, as gauged by bioluminescent imaging of live animals (Fig. 3a and Supplementary Fig. 2i) and dissected bones (Fig. 3b, c), as well as gross examination of visible tumors in the bone (Fig. 3d).

Given that bone is a frequent metastasis site for breast cancer, we performed intratibial injection of 3-month-old female *Malat1*[+/+], *Malat1*[−/−], and *Malat1*[−/−];*Malat1*[Tg/Tg] mice with the EO771 cell line, a cell line derived from a mouse mammary tumor on a C57BL/6 background[38]. Before injecting tumor cells, we conducted μCT scanning and confirmed that at this age, only female *Malat1*[−/−] mice, but not female *Malat1*[+/+] and *Malat1*[−/−];*Malat1*[Tg/Tg] mice, exhibited signs of osteoporosis (Supplementary Fig. 2j–n). After injection with $2 \times 10^5$ luciferase-labeled EO771 cells, bioluminescent signals showed no significant difference in baseline levels among the three animal groups on the injection day. At the endpoint, we observed significantly higher signals in *Malat1*[−/−] mice compared with *Malat1*[+/+] and *Malat1*[−/−];*Malat1*[Tg/Tg] mice (Fig. 3e and Supplementary Fig. 2o). After euthanasia, we collected the tibiae for ex vivo imaging (Fig. 3f, g), which confirmed in vivo imaging results. We also performed X-ray imaging of the tibiae and found that *Malat1*[−/−] mice had more osteolytic lesions (Fig. 3h). Moreover, H&E staining of bone sections demonstrated higher tumor burdens in the tibiae of *Malat1*[−/−] mice, as evidenced by more cancerous lesions in the cortical bone near the growth plate and deeper extension of tumor areas into the distal bone marrow cavity (Fig. 3i). Immunohistochemical staining of RFP (co-expressed with luciferase) supported the histologic analysis (Fig. 3i). In addition, TRAP staining revealed elevated osteoclast numbers in the tibiae of *Malat1*[−/−] mice compared with *Malat1*[+/+] and *Malat1*[−/−];*Malat1*[Tg/Tg] mice (Fig. 3j–l). Taken together with the results from the B16F1 model, these findings collectively suggest that loss of Malat1 in host mice exacerbates metastatic bone colonization by melanoma and breast cancer cells.

Because bone homeostasis is maintained by osteoclastic bone resorption and osteoblastic bone formation, we next determined whether Malat1 modulates the number and differentiation potential of osteoblasts. To this end, we stained bone sections with toluidine blue[39] (Supplementary Fig. 3a), which revealed no significant difference in the numbers of osteoblasts per bone perimeter (N.Ob/B.Pm) among *Malat1*[+/+], *Malat1*[−/−], and *Malat1*[−/−];*Malat1*[Tg/Tg] mice (Supplementary Fig. 3a, b). Further, we isolated mouse mesenchymal stem cells (MSCs) from these three groups and cultured them in

osteogenic differentiation medium[40]; we observed comparable osteogenic differentiation among all groups, as gauged by calcium mineralization (via alizarin red staining, Supplementary Fig. 3c) and alkaline phosphatase (ALP, Supplementary Fig. 3d, e). Moreover, we found no significant difference in bone formation rates among *Malat1*[+/+], *Malat1*[−/−], and *Malat1*[−/−];*Malat1*[Tg/Tg] mice, as gauged by dynamic histomorphometry measurements through sequential labeling with calcein, a fluorescent chromophore that binds to calcified skeletal structures[41,42] (Supplementary Fig. 3f, g). Taken together, our results suggest that Malat1 inhibits osteoclast differentiation and protects against osteoporosis and bone metastasis without affecting osteoblastic bone formation.

## Single-cell transcriptome analysis of bone tissues from patients with osteoporosis, osteosarcoma, or breast cancer bone metastasis

To assess the clinical relevance of MALAT1 in osteoporosis and bone metastasis, we analyzed single-cell RNA-seq data from human bone tissues. The datasets included GSE190772 with samples from two patients with breast cancer bone metastases[38,43], GSE162454 with samples from six osteosarcoma patients[44,45], and GSE169396 featuring bone tissues from a non-osteoporotic individual and three osteoporosis patients (femoral head collected during hip replacement surgery)[46]. Osteosarcomas and breast cancer bone metastases often exhibit osteolytic features.

We used the "Harmony" method[47] to remove batch effects between samples, subsequently applying dimensionality reduction to annotate cell types based on marker genes (Supplementary Fig. 4a–c). These analyses defined the cell cluster-specific transcriptome of different patient groups. We then analyzed the expression of MALAT1 in pre-osteoclasts (including monocytes and macrophages) and mature osteoclasts of the non-osteoporotic individual (Fig. 4a, b), osteoporosis patients (Fig. 4c, d), osteosarcoma patients (Fig. 4e, f), and patients with breast cancer bone metastases (Fig. 4g, h). Within each group, MALAT1 expression levels were significantly lower in osteoclasts compared with pre-osteoclasts (Fig. 4b, d, f, h). Moreover, across the four patient groups, MALAT1 expression levels in pre-osteoclasts and osteoclasts were significantly lower in patients with osteoporosis, osteosarcoma, or breast cancer bone metastasis than in the non-osteoporotic individual (Fig. 4i–k). These findings indicate that reduced MALAT1 expression in the osteoclast lineage is associated with osteoporosis and bone lesions, including breast cancer metastases and osteosarcomas.

## Malat1 deficiency promotes osteoclastogenesis through the activation of Nfatc1

Because the *Malat1*[−/−] and *Malat1*[Tg/Tg] animals used in our study are whole-body knockout and transgenic mice, the osteoporotic phenotype observed above may or may not be a direct effect of Malat1 loss in osteoclast precursors. To address this issue, we isolated primary bone

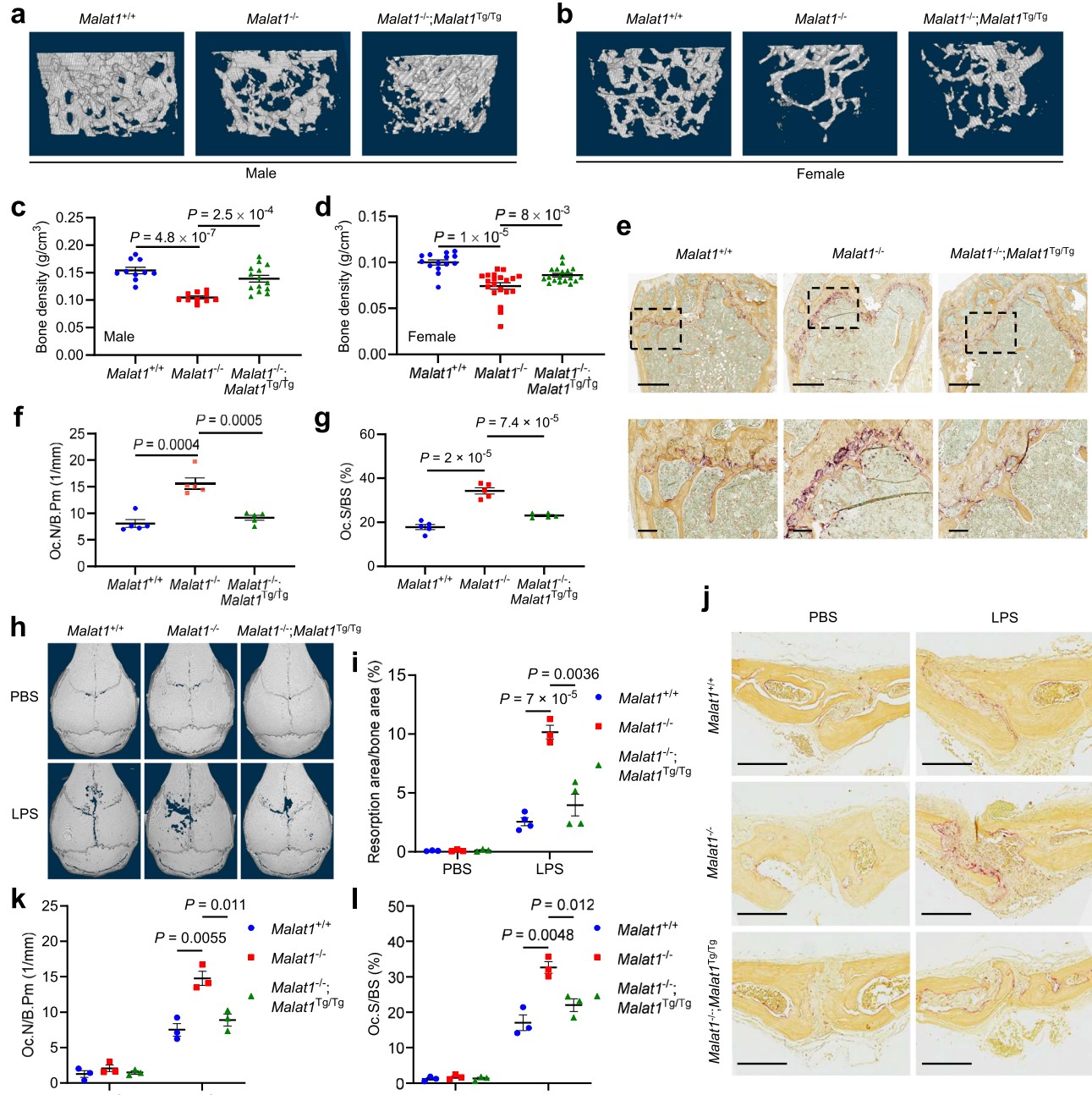

**Fig. 2 | Malat1 protects against osteoporosis. a, b** Representative µCT images of 3D bone structures of the femurs from 6-month-old male (**a**) and female (**b**) *Malat1*^(+/+), *Malat1*^(−/−), and *Malat1*^(−/−);*Malat1*^(Tg/Tg) mice. **c, d** µCT-based measurements of the bone mineral density of the femurs from 6-month-old male (**c**; *n* = 5, 5, and 7 mice per group) and female (**d**; *n* = 7, 10, and 9 mice per group) *Malat1*^(+/+), *Malat1*^(−/−), and *Malat1*^(−/−);*Malat1*^(Tg/Tg) mice, with left and right femurs for each mouse measured. **e** Representative TRAP staining images of the femurs from 6-month-old male *Malat1*^(+/+), *Malat1*^(−/−), and *Malat1*^(−/−);*Malat1*^(Tg/Tg) mice. Scale bars, 700 µm in upper panels and 100 µm in lower panels. **f, g** Quantification of osteoclast numbers per bone perimeter (Oc.N/B.Pm, **f**) and osteoclast surface per bone surface

(Oc.S/BS, **g**) in femurs of the mice described in **e**. *n* = 5 mice per group. **h–l** µCT images of the surface of calvariae (**h**), quantification of the relative resorption area (**i**), TRAP staining images of calvarial sections (**j**), the number of osteoclasts per bone perimeter (Oc.N/B.Pm, **k**), and osteoclast surface per bone surface (Oc.S/BS, **l**) in the calvarial bones from 8-week-old female *Malat1*^(+/+), *Malat1*^(−/−), and *Malat1*^(−/−);*Malat1*^(Tg/Tg) mice after the administration of PBS or LPS to the calvarial periosteum for 5 days. *n* = 3 mice per PBS group, and *n* = 4, 3, and 4 mice per LPS group in **i**. *n* = 3 mice per group in **k** and **l**. Scale bars in **j**, 200 µm. Statistical significance in **c, d, f, g, i, k,** and **l** was determined by a two-tailed unpaired *t* test. Error bars are s.e.m. Source data are provided as a Source Data file.

marrow-derived macrophages (BMMs) from *Malat1*^(+/+), *Malat1*^(−/−), and *Malat1*^(−/−);*Malat1*^(Tg/Tg) mice, and then treated these osteoclast precursors with M-CSF and RANKL for 4–6 days to induce their differentiation into osteoclasts. Genetic ablation and restoration of Malat1 expression in BMMs were confirmed by qPCR (Fig. 5a). After M-CSF- and RANKL-induced differentiation, we detected osteoclasts by TRAP staining, finding that knockout of Malat1 led to a prominent increase in

the number of TRAP-positive multinucleated osteoclasts, and that re-expression of Malat1 reversed the observed induction of osteoclastogenesis (Fig. 5b). The mRNA levels of osteoclast markers *Ctsk* and *Trap5* were much higher in *Malat1*^(−/−) cells than in *Malat1*^(+/+) and *Malat1*^(−/−);*Malat1*^(Tg/Tg) cells after RANKL treatment (Fig. 5c, d).

We also used CRISPR interference (CRISPRi) to knockdown Malat1 in cell lines. Eleven single guide RNAs (sgRNAs) targeting mouse

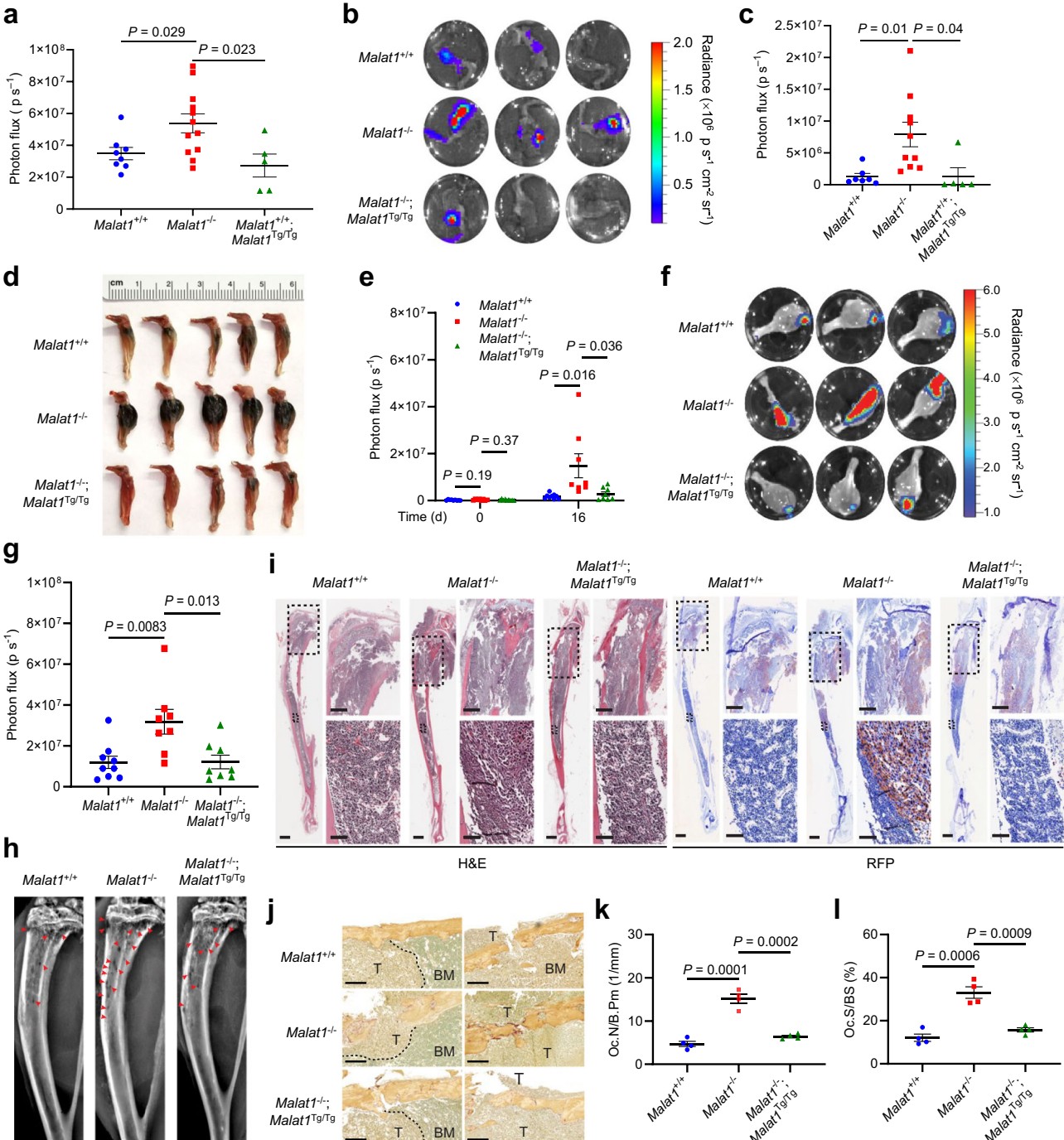

**Fig. 3 | Malat1 protects against bone metastasis of melanoma and mammary tumor cells. a** 6-month-old *Malat1*[+/+], *Malat1*[−/−], and *Malat1*[−/−];*Malat1*[Tg/Tg] male mice received intratibial injection of 5000 B16F1 melanoma cells. Bioluminescent imaging of live animals was performed at the indicated times. *n* = 8, 12, and 5 mice per group. **b–d** Bioluminescent imaging (**b**), quantification of photon flux (**c**), and photos (**d**) of the tibiae of 6-month-old male *Malat1*[+/+], *Malat1*[−/−], and *Malat1*[−/−];*Malat1*[Tg/Tg] mice at day 26 after intratibial injection of 5000 B16F1 melanoma cells. *n* = 7, 10, and 5 mice per group. **e** 3-month-old female *Malat1*[+/+], *Malat1*[−/−], and *Malat1*[−/−];*Malat1*[Tg/Tg] mice received intratibial injection of 2 × 10[5] EO771 mammary tumor cells. Bioluminescent imaging of live animals and quantification of photon flux were performed on day 0 (*n* = 9, 8, and 9 mice per group) and day 16 (*n* = 9, 8, and 8 mice per group). **f, g** Bioluminescent imaging (**f**) and quantification of photon flux (**g**) of the tibiae of 3-month-old female *Malat1*[+/+], *Malat1*[−/−],

and *Malat1*[−/−];*Malat1*[Tg/Tg] mice at day 16 after intratibial injection of 2 × 10[5] EO771 cells. *n* = 9, 8, and 8 mice per group. **h** Representative X-ray images of the tibiae of the mice described in **f**. The red arrowheads indicate osteolytic lesions appearing as tiny "holes" in the bone cavity. **i** H&E staining and immunohistochemical staining of RFP of the tibiae of the mice described in **f**. Scale bars, 3 mm (left panels); 700 μm (upper right panels); and 100 μm (lower right panels). **j–l** TRAP staining of tibiae (**j**), the number of osteoclasts per bone perimeter (Oc.N/B.Pm, **k**), and osteoclast surface per bone surface (Oc.S/BS, **l**) of the mice described in **f**. The dashed lines outline the borders between tumor tissues (T) and bone marrow tissues (BM). *n* = 4 mice per group. Scale bars in **j**, 200 μm. Statistical significance in **a**, **c**, **e**, **g**, **k**, and **l** was determined by a two-tailed unpaired *t* test. Error bars are s.e.m. Source data are provided as a Source Data file.

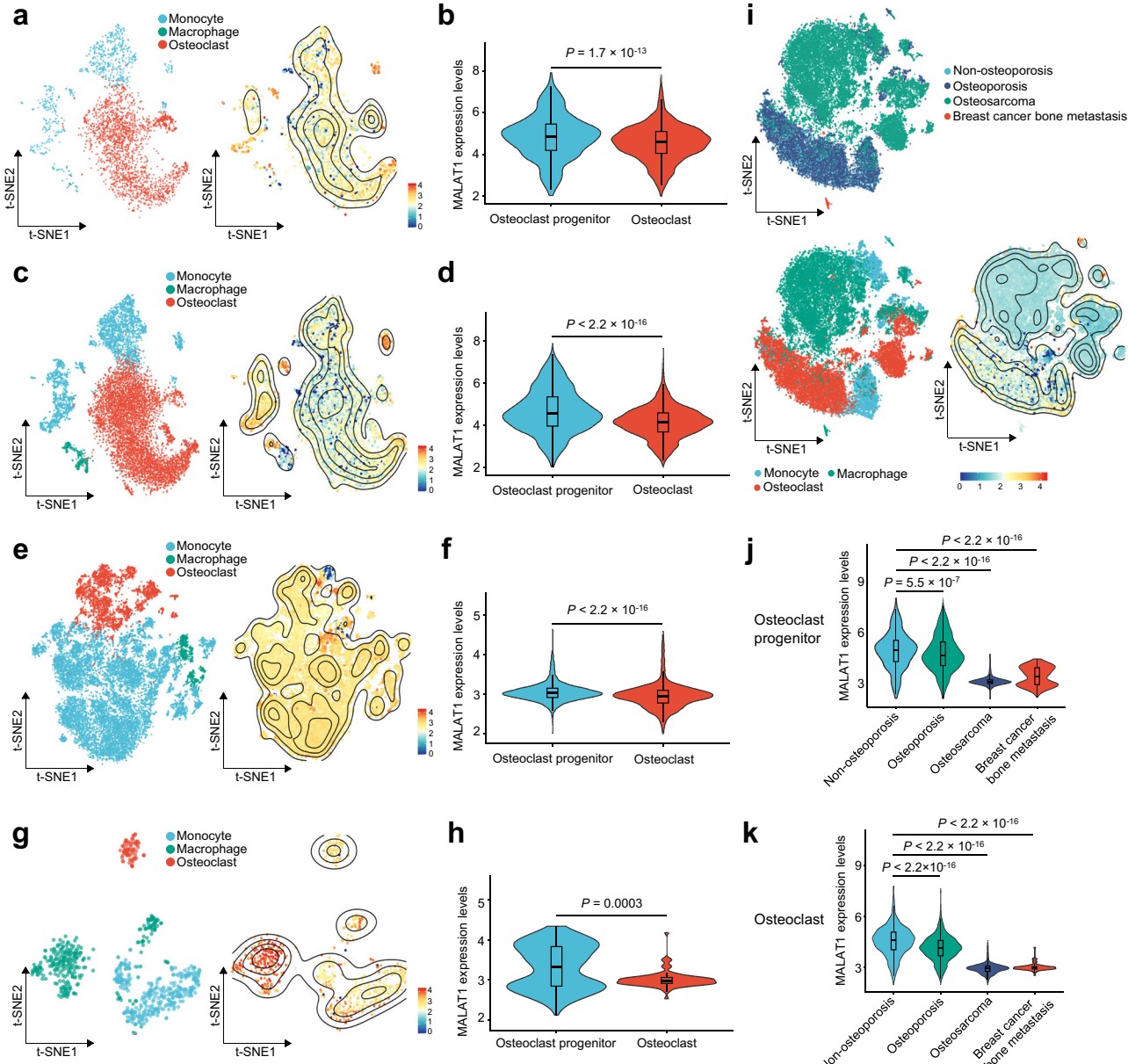

**Fig. 4 | Single-cell transcriptome analysis of bone tissues from patients with osteoporosis, osteosarcoma, or breast cancer bone metastasis. a, b** Single-cell analysis of the non-osteoporotic patient in the GSE169396 dataset ($n = 1$ patient). **a** t-SNE dimensionality reduction landscape and MALAT1 expression heatmap of monocytes, macrophages, and osteoclasts. Data represent $n = 849$ monocytes, $n = 14$ macrophages, and $n = 2217$ osteoclasts. **b** Violin plot of MALAT1 expression in osteoclast progenitors (monocytes and macrophages) and osteoclasts. **c, d** Single-cell analysis of the osteoporosis patient group in the GSE169396 dataset ($n = 3$ patients). **c** t-SNE dimensionality reduction landscape and MALAT1 expression heatmap of monocytes, macrophages, and osteoclasts. Data represent $n = 3704$ monocytes, $n = 462$ macrophages, and $n = 6551$ osteoclasts. **d** Violin plot of MALAT1 expression in osteoclast progenitors (monocytes and macrophages) and osteoclasts. **e, f** Single-cell analysis of the osteosarcoma patient group in the GSE162454 dataset ($n = 6$ patients). **e** t-SNE dimensionality reduction landscape and MALAT1 expression heatmap of monocytes, macrophages, and osteoclasts. Data represent $n = 894$ monocytes, $n = 15,283$ macrophages, and $n = 4129$ osteoclasts. **f** Violin plot of MALAT1 expression in osteoclast progenitors (monocytes and macrophages) and osteoclasts. **g, h** Single-

cell analysis of the breast cancer bone metastasis patient group in the GSE190772 dataset ($n = 4$ samples from 2 patients). **g** t-SNE dimensionality reduction landscape and MALAT1 expression heatmap of monocytes, macrophages, and osteoclasts. Data represent $n = 327$ monocytes, $n = 296$ macrophages, and $n = 80$ osteoclasts. **h** Violin plot of MALAT1 expression in osteoclast progenitors (monocytes and macrophages) and osteoclasts. **i–k** Single-cell RNA-seq meta-analysis of GSE169396, GSE162454, and GSE190772 ($n = 1$ non-osteoporosis patient, 3 osteoporosis patients, 6 osteosarcoma patients, and 4 samples from 2 breast cancer bone metastasis patients). **i** t-SNE dimensionality reduction landscape and MALAT1 expression heatmap of macrophages, monocytes, and osteoclasts from the above datasets. Data represent $n = 5774$ monocytes, $n = 16,055$ macrophages, and $n = 12,977$ osteoclasts. **j, k** Violin plots of MALAT1 expression in osteoclast progenitors (**j**) and osteoclasts (**k**) across patients with non-osteoporosis, osteoporosis, osteosarcoma, and breast cancer bone metastasis. Statistical significance in **b**, **d**, **f**, **h**, **j**, and **k** was determined by the Wilcoxon rank-sum test. The center line of the boxplot depicts the median, bounded by the inter-quartile range (IQR), 25th to 75th percentile, and the whisker represents 1.5 × IQR.

*Malat1* were tested by using the mouse B16F1 cell line (Supplementary Fig. 5a). sgRNA-2 and sgRNA-3 were chosen to knockdown Malat1 in RAW264.7 cells, which was validated by qPCR (Fig. 5e), and the two resulting Malat1-knockdown stable cell lines were named Malat1[KD1] and

Malat1[KD2]. After RANKL-induced differentiation, both Malat1[KD1] and Malat1[KD2] cells gave rise to more TRAP-positive multinucleated osteo-clasts than the control RAW264.7 cells (Fig. 5f). Fluorescent staining of F-actin rings (microfilament structures that are characteristic of

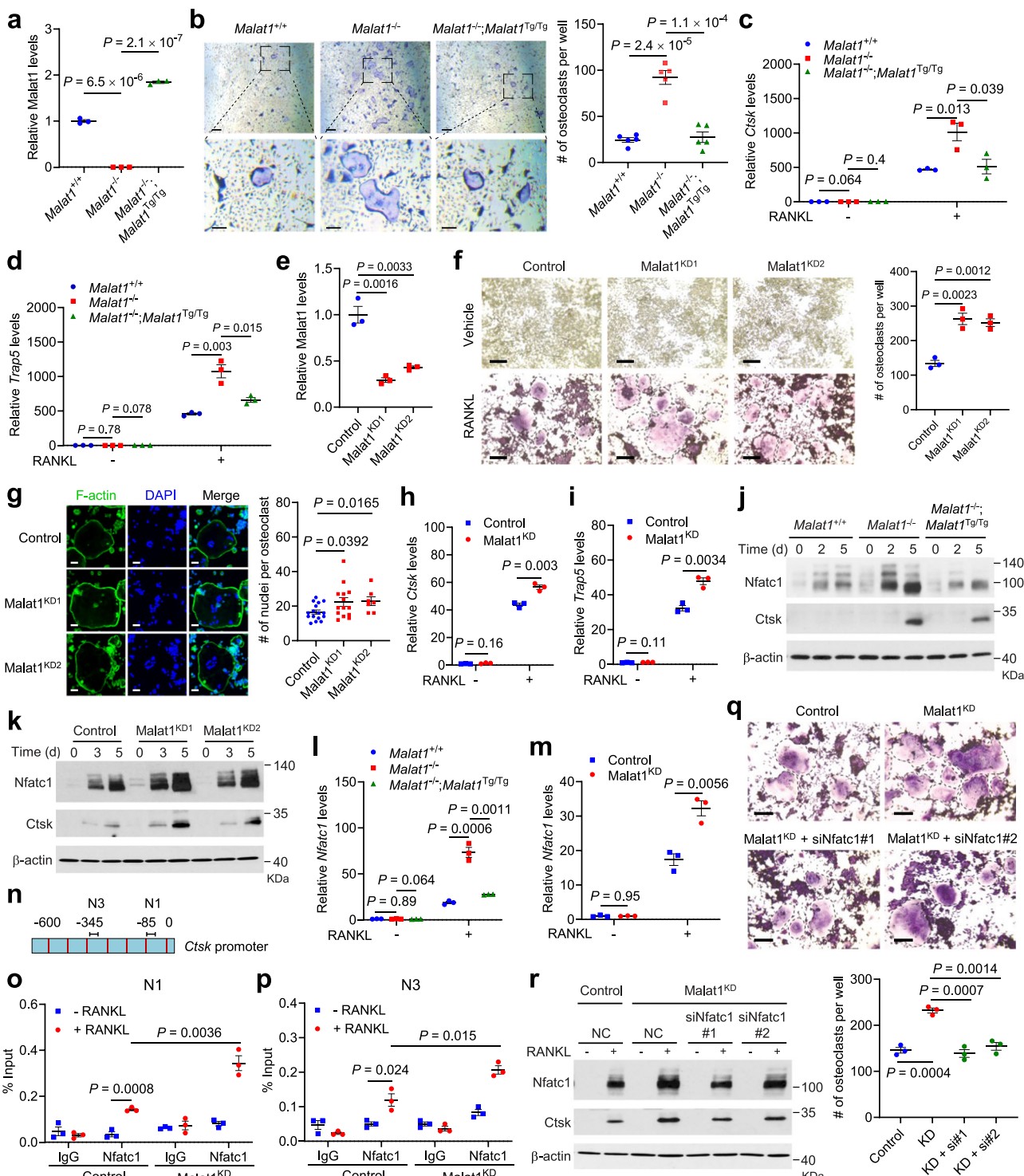

mature osteoclasts[48,49]) and nuclei, by phalloidin and DAPI, respectively, revealed that Malat1^KD1 and Malat1^KD2 cells had higher numbers of nuclei per osteoclast than the control cells (Fig. 5g). Moreover, *Ctsk* and *Trap5* mRNA levels were upregulated by knockdown of Malat1 in RANKL-treated RAW264.7 cells (Fig. 5h, i). Collectively, the results from primary BMMs and RAW264.7 cells suggest that Malat1 deficiency in osteoclast precursors promotes RANKL-induced osteoclastogenesis.

Upon binding to the RANK receptor, RANKL stimulates multiple signaling cascades, including nuclear factor-κB (NF-κB) signaling, mitogen-activated protein kinase (MAPK) signaling, and activator protein-1 (AP-1, whose major components are c-Jun and c-Fos proteins) signaling, leading to activation of downstream transcription factors,

such as Nfatc1, Mitf, and Creb1[14]. To understand how Malat1 inhibits osteoclastogenesis, we first stimulated BMMs with RANKL for short periods (5–60 min) and examined the phosphorylation events in the signaling pathways mentioned above, finding no substantial difference in the phosphorylation levels of p65 (also known as RelA), Erk1/2, Jnk, or c-Jun among the BMMs isolated from *Malat1^+/+*, *Malat1^−/−*, and *Malat1^−/−;Malat1*^Tg/Tg mice (Supplementary Fig. 5b). Thus, Malat1 loss does not affect the early kinase signaling events during RANKL-induced osteoclast differentiation.

Next, we extended the RANKL stimulation time to 2 days and 5 days, finding that Malat1 deficiency did not affect the levels of Mitf, Erk1/2, c-Fos, IκBα, Creb1, and p38 (Supplementary Fig. 5c). On the

**Fig. 5 | Malat1 deficiency promotes osteoclastogenesis through Nfatc1. a** qPCR of Malat1 in BMMs from *Malat1*[+/+], *Malat1*[−/−], and *Malat1*[−/−];*Malat1*[Tg/Tg] mice. **b** TRAP staining (left) and quantification (right) of *Malat1*[+/+], *Malat1*[−/−], and *Malat1*[−/−];*Malat1*[Tg/Tg] BMMs treated with M-CSF and RANKL. Scale bars, 125 μm (upper) and 50 μm (lower). *n* = 5 wells per group. **c, d** qPCR of *Ctsk* (**c**) and *Trap5* (**d**) in the BMMs described in **b**. **e** qPCR of Malat1 in Malat1-knockdown RAW264.7 cells. **f** TRAP staining (left) and quantification (right) of RANKL-treated control and Malat1-knockdown RAW264.7 cells. Scale bars, 100 μm. *n* = 3 wells per group. **g** Left: RANKL-treated control and Malat1-knockdown RAW264.7 cells were stained with Phalloidin Green 488 and DAPI. Right: data quantification. Scale bars, 50 μm. *n* = 16, 14, and 7 cells per group. **h, i** qPCR of *Ctsk* (**h**) and *Trap5* (**i**) in RANKL-treated control and Malat1-knockdown RAW264.7 cells. **j, k** Immunoblotting of Nfatc1, Ctsk, and β-actin in *Malat1*[+/+], *Malat1*[−/−], and *Malat1*[−/−];*Malat1*[Tg/Tg] BMMs treated with M-CSF and RANKL (**j**), and in RANKL-treated control and Malat1-knockdown RAW264.7 cells (**k**). **l, m** qPCR of *Nfatc1* in *Malat1*[+/+], *Malat1*[−/−], and *Malat1*[−/−];*Malat1*[Tg/Tg] BMMs treated with M-CSF and RANKL (**l**), and in RANKL-treated control and Malat1-knockdown RAW264.7 cells (**m**). **n** The mouse *Ctsk* promoter. Primers previously reported for amplifying N1 and N3 regions were used for ChIP-qPCR. **o, p** ChIP-qPCR showing the occupancy of the N1 (**o**) and N3 (**p**) regions of the *Ctsk* promoter by Nfatc1 immunoprecipitated from RANKL-treated control or Malat1-knockdown RAW264.7 cells. **q, r** Control and Malat1-knockdown RAW264.7 cells were transfected with negative control (NC) or Nfatc1 siRNA. After 24 h, the cells were treated with RANKL for 5 days, followed by TRAP staining and quantification (**q**). Scale bars, 100 μm. Cell lysates were subjected to immunoblotting of Nfatc1, Ctsk, and β-actin (**r**). *n* = 3 wells per group. Statistical significance in **a–i**, **l**, **m**, and **o–q** was determined by a two-tailed unpaired *t* test. Error bars are s.e.m. *n* = 3 biological replicates in **a**, **c–e**, **h**, **i**, **l**, **m**, **o**, and **p**. The experiments in **j**, **k**, and **r** were repeated independently three times, yielding similar results. Source data are provided as a Source Data file.

other hand, compared with *Malat1*[+/+] and *Malat1*[−/−];*Malat1*[Tg/Tg] BMMs, Malat1-knockout BMMs showed more induction of Nfatc1 and its transcriptional target Ctsk[50] (Fig. 5j). We observed similar results from RANKL-treated Malat1[KD1] and Malat1[KD2] RAW264.7 cells (Fig. 5k), and these cells exhibited increased accumulation of Nfatc1 both in the nucleus and in the cytoplasm (Supplementary Fig. 5d). Nfatc1 is known to act as its own transcription factor[11,51]. Consistently, Malat1-knockout BMMs and Malat1-knockdown RAW264.7 cells showed more induction of *Nfatc1* mRNA levels after RANKL stimulation, compared with their Malat1 wild-type counterparts (Fig. 5l, m). *Ctsk*, a classic Nfatc1 target gene, contains two Nfatc1-binding sites in the promoter region[52] (Fig. 5n). Chromatin immunoprecipitation-qPCR assays revealed that after RANKL treatment, Malat1-knockdown RAW264.7 cells showed more occupancy of these two regions by Nfatc1 than the control RAW264.7 cells (Fig. 5o, p). Similar results were also observed on Nfatc1 occupancy of promoter regions of other target genes, *Nfatc1* and *Acp5* (Supplementary Fig. 5e, f). Importantly, knockdown of Nfatc1 in Malat1-depleted RAW264.7 cells reversed the induction of osteoclastogenesis and Ctsk expression upon stimulation with RANKL (Fig. 5q, r and Supplementary Fig. 5g). Taken together, these results suggest that Malat1 loss promotes osteoclast differentiation through Nfatc1.

We also sought to determine the effect of Malat1 overexpression. Consistent with Malat1 being a highly abundant lncRNA, we tested various methods and could only overexpress Malat1 in RAW264.7 cells by using the piggyBac transposon system and electroporation. The resulting overexpression level was approximately a 1.7-fold increase over the endogenous expression level (Supplementary Fig. 6a), which did not lead to significant differences in RANKL-induced osteoclastogenesis or the expression of Nfatc1, Trap5, and Ctsk (Supplementary Fig. 6b–g). Moreover, BMMs from *Malat1*[Tg/Tg] mice exhibited approximately a 1.5-fold increase in Malat1 expression relative to BMMs from *Malat1*[+/+] mice (Supplementary Fig. 6h). Compared with *Malat1*[+/+] mice, *Malat1*[Tg/Tg] mice did not display any significant difference in bone density or other bone parameters based on μCT analysis (Supplementary Fig. 6i–m). The challenge of achieving substantial Malat1 overexpression in wild-type cells and mice limited a comprehensive examination of its overexpression effects. However, considering that reduced MALAT1 expression in pre-osteoclasts and osteoclasts is associated with osteoporosis and bone metastasis (Fig. 4i–k), our loss-of-function approach, coupled with re-expression of Malat1 in Malat1-deficient mice and cell lines, is suitable for this investigation.

To further extend our study to human osteoclastogenesis, we treated the U937 human pre-osteoclast/monocyte cell line with phorbol 12-myristate 13-acetate (PMA, 100 ng/mL) for 2 days, followed by 12–14 days of human M-CSF (50 ng/mL) and RANKL (100 ng/mL) treatment, as described previously[53,54]. NFATC1 expression showed an initial upregulation in the first 5 days, followed by a decrease, while the osteoclast marker TRAP exhibited a progressive elevation (Supplementary Fig. 7a). To determine the role of MALAT1 in human osteoclast differentiation, we used CRISPRi to knockdown MALAT1. Five sgRNAs (sg1–5) that target the human *MALAT1* promoter were tested in HEK293T cells, and four out of five sgRNAs showed ~50% knockdown efficiency (Supplementary Fig. 7b). We used sg2 and sg4 to deplete MALAT1 in U937 cells, achieving over 95% knockdown efficiency in this cell line (Supplementary Fig. 7c). Subsequently, osteoclastogenesis assays revealed a higher number of TRAP-positive osteoclasts in the MALAT1 knockdown group compared with the control group (Supplementary Fig. 7d, e). Moreover, NFATC1 and TRAP expression levels were elevated in MALAT1-knockdown U937 cells during osteoclast differentiation (Supplementary Fig. 7f–h). Hence, MALAT1 functions as a suppressor of both mouse and human osteoclastogenesis.

## Malat1 binds Tead3 to inhibit Nfatc1 activity and osteoclastogenesis

How does Malat1 regulate Nfatc1? The binding of Nfatc1 to other proteins can lead to either activation or inhibition of the transcriptional activity of Nfatc1[55], while lncRNAs often exert their functions by interacting with proteins, and this mode of action has been demonstrated for Malat1[15,17,22,56]. We speculated that Malat1 might regulate the Nfatc1 auto-amplification loop by interacting with Nfatc1 and/or its binding proteins, and thus we searched a database of protein-protein interactions, Mentha (http://mentha.uniroma2.it/index.php). Of all potential NFATC1-interacting proteins (Supplementary Fig. 8a), TEAD was of particular interest, because our previous chromatin isolation by RNA purification coupled to mass spectrometry (ChIRP-MS) analysis captured an endogenous Malat1-Tead interaction in mouse tissues, which was validated by ChIRP-Western, RNA pulldown, and RNA immunoprecipitation (RIP) assays[22]. Therefore, we hypothesized that Malat1 might regulate Nfatc1 through Tead.

We examined the protein levels of the four Tead family members (Tead1–4) in BMMs and RAW264.7 cells, along with several other mouse cell lines. Interestingly, Tead1 and the Tead co-activator Yap were undetectable in BMMs and RAW264.7 cells but were abundantly expressed in the mouse melanoma cell line B16F1, mouse embryonic fibroblasts (MEF), MSC, and the mouse fibroblast cell line L929 (Fig. 6a). In contrast, Tead3 showed a relatively specific expression pattern in primary BMMs (Fig. 6a). To determine whether Malat1 interacts with Nfatc1 in pre-osteoclasts, we performed RIP assays, finding that Malat1 was enriched in both pan-Tead and Tead3 immunoprecipitates from RAW264.7 cells (Fig. 6b, c), which validated the interaction between Malat1 and Tead3 in these osteoclast precursors. To determine whether Malat1 directly binds to Tead3, we performed RNA pulldown assays with six non-overlapping biotinylated fragments of Malat1 (P1-P6; 1.1–1.2 kb each) generated by in vitro transcription[22], and we found that all six Malat1 fragments, but not an unrelated nuclear RNA U1, bound to Tead3 protein (Fig. 6d), suggesting that the

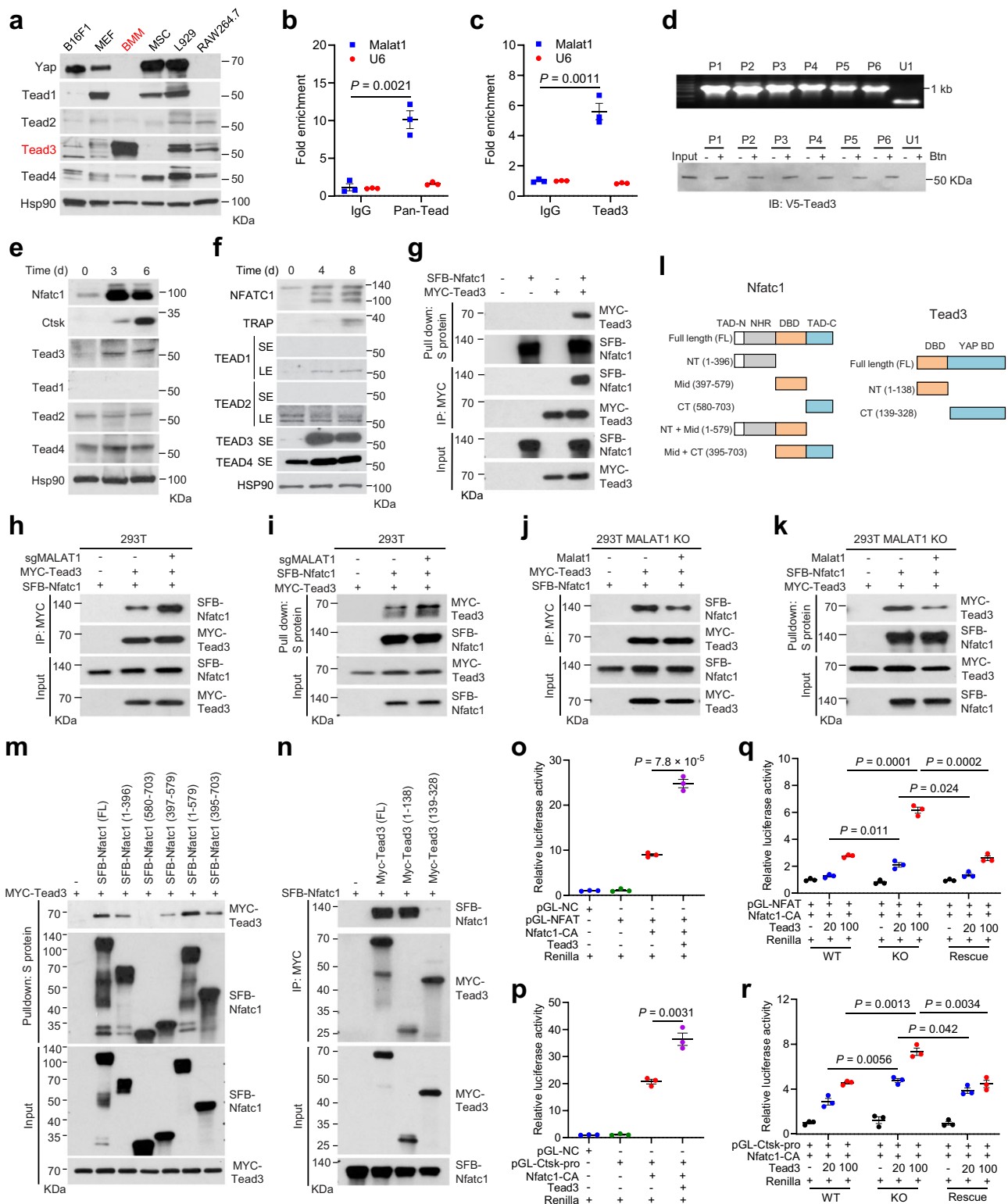

Tead3-binding sites may be distributed diffusely on Malat1. Consistent with this, ectopic expression of each of the six Malat1 fragments in MALAT1-depleted U937 cells partially reversed, while re-expression of full-length Malat1 completely reversed RANKL-induced osteoclastogenesis (Supplementary Fig. 8b–d).

Interestingly, RANKL treatment of RAW264.7 and U937 cells upregulated Tead3, but not other Tead family members (Fig. 6e, f). Co-immunoprecipitation (co-IP) assays revealed that Tead3, but not Yap, interacted with Nfatc1 (Fig. 6g and Supplementary Fig. 8e). After

validating the interaction of Tead3 with Malat1 and Nfatc1, we sought to determine whether Malat1 modulates the binding of Tead3 to Nfatc1. To this end, we generated MALAT1-knockout HEK293T cells (Supplementary Fig. 9a, b) and transfected these cells with Tead3 and Nfatc1. Co-IP assays showed that Malat1 loss significantly increased the interaction between Tead3 and Nfatc1 (Fig. 6h, i). To further corroborate this result, we re-expressed Malat1 in MALAT1-knockout HEK293T cells (Supplementary Fig. 9c), finding that restoring Malat1 expression reduced the Tead3-Nfatc1 interaction (Fig. 6j, k).

**Fig. 6 | Malat1 binds to Tead3 to inhibit Nfatc1 activity. a** Immunoblotting of Yap and Tead1-4 in B16F1, MEF, BMM, MSC, L929, and RAW264.7 cells. **b, c** Total Tead (**b**) or Tead3 (**c**) was immunoprecipitated from RAW264.7 cells. Tead- or Tead3-bound Malat1 was quantitated by qPCR. **d** Biotinylated (Btn) Malat1 fragments were synthesized in vitro (upper), incubated with V5-Tead3-overexpressing HEK293T cell lysates, and pulled down with streptavidin beads. The bound proteins were immunoblotted with a V5-specific antibody (lower). **e, f** Immunoblotting of NFATC1, Ctsk or TRAP, and TEAD1-4 in RANKL-treated RAW264.7 (**e**) or U937 (**f**) cells. SE short exposure, LE long exposure. **g** HEK293T cells were co-transfected with SFB-Nfatc1 and MYC-Tead3, followed by pulldown with S-protein beads or a MYC-specific antibody and immunoblotting with antibodies against MYC and FLAG. **h, i** Control and MALAT1-knockout HEK293T cells were co-transfected with SFB-Nfatc1 and MYC-Tead3, followed by pulldown with a MYC-specific antibody (**h**) or S-protein beads (**i**) and immunoblotting with antibodies against MYC and FLAG. **j, k** MALAT1-knockout and Malat1-restored HEK293T cells were co-transfected with SFB-Nfatc1 and MYC-Tead3, followed by pulldown with a MYC-specific antibody (**j**) or S-protein beads (**k**) and immunoblotting with antibodies against MYC and FLAG.

**l** Mouse Nfatc1, Tead3, and truncation mutants. **m** HEK293T cells were co-transfected with MYC-Tead3 and SFB-Nfatc1 (full-length or truncated), followed by pulldown with S-protein beads and immunoblotting with antibodies against MYC and FLAG. **n** HEK293T cells were co-transfected with SFB-Nfatc1 and MYC-Tead3 (full-length or truncated), followed by pulldown with a MYC-specific antibody and immunoblotting with antibodies against MYC and FLAG. **o, p** Luciferase activity in HEK293T cells co-transfected with Tead3, constitutively active Nfatc1 (Nfatc1-CA), Renilla luciferase, and a firefly luciferase reporter containing tandem Nfatc1-binding sites (**o**) or the *Ctsk* promoter (**p**). **q, r** Luciferase activity in wild-type, MALAT1-knockout, and Malat1-restored HEK293T cells co-transfected with Tead3, Nfatc1-CA, Renilla luciferase, and a firefly luciferase reporter containing tandem Nfatc1-binding sites (**q**) or the *Ctsk* promoter (**r**). Statistical significance in **b**, **c**, and **o**–**r** was determined by a two-tailed unpaired *t* test. Error bars are s.e.m. $n = 3$ biological replicates. The experiments in **a**, **d**–**k**, **m**, and **n** were repeated independently three times, yielding similar results. Source data are provided as a Source Data file.

Besides TEAD, our Mentha database search also revealed other candidate NFATC1-interacting proteins, among which FOS, JUN, and CREB1 have been reported to regulate osteoclast differentiation[57,58] (Supplementary Fig. 8a). We pulled down NFATC1 from the control, MALAT1-knockout, and Malat1-restored HEK293T cells, followed by immunoblotting with antibodies against FOS, JUN, and CREB1. While we did not detect an interaction of FOS with NFATC1, we observed interactions of JUN and CREB1 with NFATC1; however, unlike the TEAD3-NFATC1 interaction, these interactions were not affected by MALAT1 (Supplementary Fig. 9d, e).

Nfatc1 protein contains four domains: two transcription activation domains (TAD) in N-terminal and C-terminal regions, a central DNA-binding domain (DBD), and an N-terminal regulatory domain (NHR)[55] (Fig. 6l). Tead3 protein mainly consists of two domains: an N-terminal DBD (also known as the TEA domain) and a C-terminal YAP-binding domain[59]. Accordingly, we generated truncation mutants of Nfatc1 and Tead3 (Fig. 6l) and performed co-IP assays, finding that both the N-terminal region (containing a TAD and the NHR domain) and the central DBD, but not the C-terminal TAD of Nfact1, could bind Tead3 (Fig. 6m). In addition, co-IP assays using truncated Tead3 mutants and full-length Nfatc1 demonstrated that the TEA domain of Tead3, but not the YAP-binding domain, was responsible for interaction with Nfatc1 (Fig. 6n).

We further examined whether Malat1 and Tead3 modulate Nfatc1's transcriptional activity by using a luciferase reporter containing either tandem Nfatc1-binding sites or the *Ctsk* promoter, and we found that overexpression of Tead3 indeed enhanced the transcriptional activity of Nfatc1 (Fig. 6o, p). We then transfected Tead3 into wild-type, MALAT1-knockout, and Malat1-restored HEK293T cells, finding that Tead3 overexpression led to higher dose-dependent increases in Nfatc1 activity in MALAT1-knockout cells compared with either wild-type or Malat1-rescued cells (Fig. 6q, r). These results lend support to a model in which Malat1 loss derepresses Tead3, which in turn binds and activates Nfatc1.

Does Tead3 mediate the role of Malat1 in osteoclastogenesis? By using the CRISPR activation (CRISPRa) method to activate endogenous Tead3 expression in RAW264.7 cells (Fig. 7a, b), we found that overexpression of Tead3 promoted osteoclast differentiation (Fig. 7c, d) and upregulated Nfatc1, Trap5, and Ctsk expression (Fig. 7e–h). Next, we knocked down Tead3 in RAW264.7 cells (Fig. 7i), finding that Tead3 depletion attenuated RANKL-induced osteoclast differentiation (Fig. 7j, k) and downregulated the expression of Nfatc1 and Ctsk (Fig. 7l). Moreover, silencing Tead3 in Malat1-depleted RAW264.7 cells counteracted RANKL-mediated induction of osteoclastogenesis and expression of Nfatc1 and Ctsk (Fig. 7m–p). Therefore, Malat1 loss promotes osteoclast differentiation in a Tead3- and Nfatc1-dependent manner.

Similar to RAW264.7 cells (Fig. 6e), during osteoclast differentiation of the human pre-osteoclast cell line U937, TEAD3 was also the most upregulated TEAD family member (Fig. 6f). Thus, we examined TEAD3's function in U937 cells, finding that shRNA-mediated knockdown of TEAD3 impaired human osteoclastogenesis (Supplementary Fig. 10a–c) and downregulated the expression of NFATC1 and TRAP at both mRNA and protein levels (Supplementary Fig. 10d–f). It should be noted that MALAT1 depletion did not affect the upregulation of TEAD3 during osteoclast differentiation (Supplementary Fig. 10g), suggesting that MALAT1 does not regulate TEAD3's expression levels (but instead inhibits the TEAD3-NFATC1 interaction). Taken together with the results from RAW264.7 cells, these findings collectively suggest that TEAD3 promotes both mouse and human osteoclastogenesis.

## Discussion

This study identified Malat1 as an osteoporosis-suppressing and bone metastasis-inhibiting lncRNA that is downregulated during RANKL-triggered osteoclastogenesis. RANKL stimulates multiple signaling pathways, most of which (such as MAPK and NF-κB pathways) can also be activated by other factors, and yet RANKL is indispensable and irreplaceable in osteoclastogenesis[14], which could be explained by Nfatc1's role as a specific master regulator of osteoclast differentiation. As a transcriptional factor of its own coding gene and other osteoclast-specific genes, the binding of Nfatc1 to other nuclear proteins can lead to synergistic activation of gene transcription[55], as exemplified by the AP-1 transcription factor complex, which interacts with Nfatc1 to boost the transcriptional activity of Nfatc1[60]. Here, we identified Tead3 as a macrophage-osteoclast–specific Tead family member and a binding partner of Nfatc1, and our data suggest a model (Fig. 8) in which Malat1 binds and sequesters Tead3, blocking Tead3 from associating with Nfatc1 and inducing the transcription of Nfatc1 target genes, including *Nfatc1* itself and *Ctsk*. In response to RANKL stimulation, downregulation of Malat1 releases Tead3, thereby enhancing both the Tead3-Nfatc1 interaction as well as the transcription factor occupancy of Nfatc1 target genes, which leads to activation of Nfatc1-mediated gene transcription and osteoclast differentiation.

In addition to hyperactivation of osteoclastic bone resorption, suppression of osteoblastic bone formation can also contribute to low BMD and osteoporosis. Several previous publications reported that MALAT1 promotes osteoblast differentiation by acting as a competing endogenous RNA (ceRNA) to microRNAs (miRNAs), i.e., a "miRNA sponge", based on MALAT1 shRNA or siRNA knockdown in cell culture[61–65]. How a nuclear lncRNA could bind miRNAs is unclear. Considering the pitfalls in using RNAi, large genomic deletion (*MALAT1*, a single-exon gene, is ~7 kb in mice and ~8 kb in humans), promoter deletion, and RNase H-dependent antisense oligonucleotide approaches to deplete nuclear lncRNAs[15,16,66–68], we used different

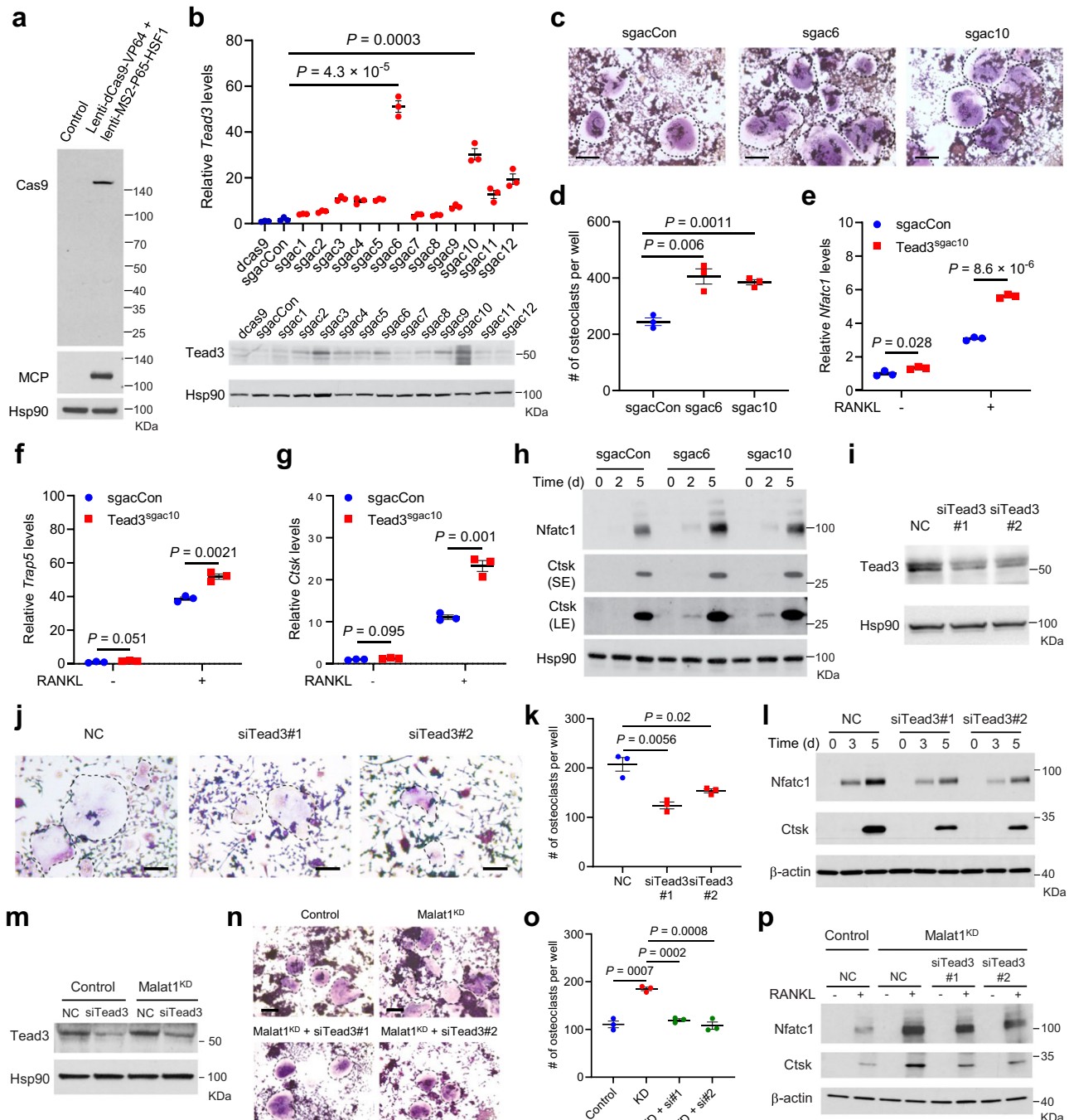

**Fig. 7 | Tead3 promotes osteoclastogenesis and mediates the effect of Malat1 deficiency. a** Immunoblotting of Cas9 and MCP in RAW264.7 cells transduced with lenti-dCas9-VP64 and lenti-MS2-P65-HSF1. **b** qPCR (upper) and immunoblotting (lower) of Tead3 in RAW264.7 cells with CRISPRa-mediated overexpression of Tead3. $n = 3$ biological replicates per group. **c, d** TRAP staining images (**c**) and quantification (**d**) of control and Tead3-overexpressing RAW264.7 cells treated with RANKL (50 ng/mL) for 5 days. Multinucleated TRAP-positive cells (outlined by dashed lines) were counted. Scale bars, 100 μm. $n = 3$ wells per group. **e–g** qPCR of *Nfatc1* (**e**), *Trap5* (**f**), and *Ctsk* (**g**) in control and Tead3-overexpressing RAW264.7 cells treated with RANKL (50 ng/mL) for 3 days. $n = 3$ biological replicates per group. **h** Immunoblotting of Nfatc1, Ctsk, and Hsp90 in control and Tead3-overexpressing RAW264.7 cells treated with RANKL (50 ng/mL) for 2 days and 5 days. **i** Immunoblotting of Tead3 and Hsp90 in RAW264.7 cells transfected with two independent Tead siRNAs or scrambled negative control (NC). **j, k** TRAP staining images (**j**) and quantification (**k**) of control and Tead3-knockdown

RAW264.7 cells treated with RANKL (50 ng/mL) for 5 days. Multinucleated TRAP-positive cells (outlined by dashed lines) were counted. Scale bars: 100 μm. $n = 3$ wells per group. **l** Immunoblotting of Nfatc1, Ctsk, and β-actin in control and Tead3-knockdown RAW264.7 cells treated with RANKL for 3 days and 5 days. **m** Immunoblotting of Tead3 and Hsp90 in control and Malat1-knockdown RAW264.7 cells transfected with Tead3 siRNA or scrambled negative control (NC). **n–p** Control and Malat1-knockdown RAW264.7 cells were transfected with Tead3 siRNA or scrambled negative control (NC). 24 h after siRNA transfection, the cells were treated with RANKL for 5 days, followed by TRAP staining (**n**) and quantification (**o**) of multinucleated TRAP-positive cells (outlined by dashed lines). Scale bars, 100 μm. Cell lysates were subjected to immunoblotting of Nfatc1, Ctsk, and β-actin (**p**). $n = 3$ wells per group in **o**. Statistical significance in **b**, **d–g**, **k**, and **o** was determined by a two-tailed unpaired *t* test. Error bars are s.e.m. The experiments in **a**, **h**, **i**, **l**, **m**, and **p** were repeated independently three times, yielding similar results. Source data are provided as a Source Data file.

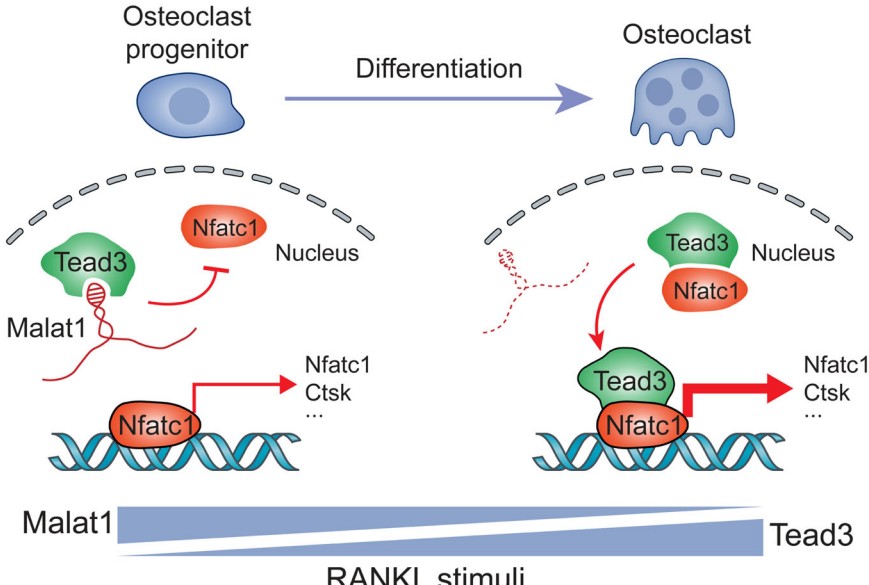

**Fig. 8 | Model for the regulation of osteoclastogenesis by the Malat1-Tead3-Nfatc1 axis.** In the presence of Malat1 lncRNA, Malat1 sequesters Tead3, impeding its interaction with Nfatc1 and consequently constraining Nfatc1's transcriptional activity. In the absence of Malat1, Tead3 is derepressed, facilitating its binding to Nfatc1 and promoting Nfatc1-mediated gene transcription. During RANKL-induced osteoclastogenesis from osteoclast progenitors (monocytes and macrophages), Malat1 is downregulated, and Tead3 is upregulated, thereby favoring osteoclast differentiation.

methods for loss-of-function analyses of Malat1 in vitro and in vivo, including CRISPRi, double gRNA-mediated focal deletion in the 5′ region (without affecting the promoter), and insertional inactivation, along with genetic rescue experiments. In the present study, loss of Malat1 in pre-osteoclasts (including RAW264.7 cells and primary BMMs from *Malat1⁻/⁻* mice) promoted osteoclastogenesis, a phenotype that could be reversed by restoration of Malat1 expression. On the other hand, the results from *Malat1⁺/⁺*, *Malat1⁻/⁻*, and *Malat1⁻/⁻;Malat1^Tg/Tg* mice, as well as osteoblast differentiation assays of MSCs isolated from these animals, showed no evidence for the regulation of osteoblastogenesis by Malat1 (Supplementary Fig. 3a–g).

Via its C-terminal YAP-binding domain, the TEAD family of transcription factors binds to the transcriptional co-activator YAP to turn on the expression of TEAD-YAP target genes[69]. In doing so, TEAD proteins and YAP are involved in several processes, including organ growth, regeneration, tumor progression, and metastasis[69]. Potentially druggable sites in the protein-protein interaction between YAP and TEAD, as well as a highly conserved palmitoylation pocket in TEADs, have been identified and exploited for drug development[70]. In an ongoing clinical trial (NCT04665206), the first-in-class TEAD inhibitor was well tolerated with durable antitumor responses in patients with advanced mesothelioma or other cancers harboring *NF2* mutations. However, whether the TEAD family can function in a YAP-independent manner is elusive. In this study, Tead3, but not other Tead family members, exhibited a specific expression pattern in primary bone marrow macrophages (pre-osteoclasts), whereas Tead1 and Yap were barely detectable in these cells (Fig. 6a). We further found that Tead3 binds and activates Nfatc1 via its N-terminal TEA domain (but not its C-terminal YAP-binding domain), which is required for RANKL-induced osteoclastogenesis, thus revealing a non-canonical function of Tead that is mediated by Nfatc1 and is controlled by Malat1 lncRNA. Our findings suggest the therapeutic potential of developing agents that disrupt the TEAD3-NFATC1 interaction for treating osteoporosis and bone metastasis. For example, the TEAD inhibitor could emerge as a drug that not only elicits antitumor responses through the tumor cell-intrinsic mechanism, but also inhibits bone metastasis through the tumor cell-extrinsic mechanism.

Future studies should address the following issues: first, we found that Malat1 expression is downregulated during osteoclast differentiation; yet, how this lncRNA is regulated by pro-osteoclastogenic factors under physiological and pathological conditions is unknown. Second, our study identified a Malat1-Tead3-Nfatc1 axis that regulates osteoclastogenesis, but it is possible that additional binding partners of Malat1 and Tead3 could also be involved in osteoclast differentiation. Third, our previous study[22] and the present study collectively demonstrate that Malat1 binds to Tead to inactivate Yap-Tead's pro-metastatic function in cancer cells and to inhibit Nfatc1-Tead3's pro-osteoclastogenic function in pre-osteoclasts, revealing both tumor-intrinsic and tumor-extrinsic mechanisms of Malat1 in metastasis suppression. Whether Malat1 suppresses metastasis at other anatomic sites (in addition to the lung and bone) and in cancer types in addition to breast cancer and melanoma warrants further investigation. Finally, whether Malat1 regulates other types of niche cells to control metastasis remains an open question.

## Methods
### Genetically engineered mouse models
All animal studies were performed in accordance with a protocol approved by the Institutional Animal Care and Use Committee of MD Anderson Cancer Center. Animals were housed at 70 °F–74 °F (set point: 72 °F) with 40–55% humidity (set point: 45%). The light cycle of animal rooms is 12 h of light and 12 h of dark. Malat1-knockout mice with targeted inactivation of *Malat1* (*Malat1⁻/⁻*) and mice with targeted transgenic expression of Malat1 from the *ROSA26* locus (*Malat1^Tg/Tg*) were described in our previous paper[22]. To restore Malat1 expression in *Malat1⁻/⁻* mice, we bred *Malat1⁻/⁻* mice to *Malat1^Tg/Tg* mice and further mated their heterozygous offspring to produce *Malat1⁻/⁻;Malat1^Tg/Tg* mice. All mice described here were on a C57BL/6 background. Primers for PCR genotyping were listed in our previous paper[22].

### LPS-induced inflammatory osteoporosis model
The procedure for inflammation-induced bone destruction was performed as previously described[36]. Briefly, 8-week-old female mice were injected above the calvarium with 12.5 mg/kg of LPS (Sigma, L4391) or vehicle (phosphate buffered saline, PBS). After 6 days, calvariae were

collected and analyzed by µCT, followed by embedding, sectioning, and TRAP staining.

## µCT-based bone scanning and analysis

Mouse femurs were scanned on a Bruker microCT SkyScan 1276 (Bruker, Kontich, Belgium) with a source voltage of 55 kV, a source current of 200 µA, a filter setting of Al 0.2 mm, and a pixel size of 13 µm at 2016 × 1344. We used 435 ms exposure time and the step and shoot mode with rotation step 0.400 degrees. Backward projection datasets of all femurs were reconstructed by using Insta-Recon software (Bruker microCT, Kontich, Belgium). Parameters for reconstruction were windowing 0–0.08 intensity, ring artifact reduction 5, beam hardening 23%, and automatic post-alignment correction. The proximal end of the femur corresponding to a 0–1.3 mm region (consisting of 100 sections over the region of interest) below the growth plate was selected and analyzed by using CTAn software (Version 1.18 8.0+, Bruker microCT, Belgium) to determine the trabecular BMD, trabecular bone volume per tissue volume (BV/TV), trabecular number (Tb.N), and trabecular thickness (Tb.th). The threshold value for µCT was set at 150–250. All calculations were performed based on 3D standard microstructural analysis. For visualizing the femurs, a 3D model was created with CTVox software (Version 3.3.0, Bruker microCT, Belgium) based on the same region of the microstructural analysis.

Mouse calvariae were scanned on a Bruker microCT SkyScan 1276 (Bruker, Kontich, Belgium) with the same parameters as those used for scanning femurs, except the exposure time (400 ms), windowing (0–0.06 intensity), ring artifact reduction (3), and beam hardening (20). For visualizing the osteolytic area of the calvaria, a 3D model was created with CTVox software (Version 3.3.0, Bruker microCT, Belgium). A region of 8 mm × 8 mm centered at the midline suture was used for further quantitative analysis with ImageJ (Version 1.53 m).

## Bone metastasis assay

B16F1 melanoma cells with stable expression of firefly luciferase (Addgene, 39196) were cultured to 70% confluence and harvested during the log phase of growth. Then, 5000 cells were resuspended in sterile PBS, and the tumor cell suspensions were injected into the left tibiae of 6-month-old male $Malat1^{+/+}$, $Malat1^{-/-}$, or $Malat1^{-/-};Malat1^{Tg/Tg}$ mice by using a 27-gauge needle under isoflurane anesthesia. Bioluminescence imaging was performed at days 0, 14, and 25 after intratibial injection under isoflurane anesthesia by using an IVIS 200 imaging platform (Perkin Elmer), following the intraperitoneal injection of 100 µl D-luciferin substrate (25 mg/mL in PBS, Perkin Elmer). The mammary tumor line EO771 was labeled with firefly luciferase and RFP and injected into the tibiae of 3-month-old female $Malat1^{+/+}$, $Malat1^{-/-}$, or $Malat1^{-/-};Malat1^{Tg/Tg}$ mice at $2 \times 10^5$ cells per mouse. Bioluminescence imaging was performed at day 0 and day 16 after intratibial injection. At the endpoint, the mice were euthanized, and the tibiae were collected for ex vivo bioluminescence imaging and photography. The imaging data were processed and quantitated with Living Image Software version 4.7 (Perkin Elmer). For detecting osteolytic lesions, the tibiae were scanned and processed with the Faxitron MX-20 Digital X-Ray System (Wheeling, IL). None of the following IACUC-approved euthanasia criteria was exceeded: (1) the maximum cumulative tumor burden of 2.0 cm in diameter; (2) the tumor impedes eating, urination, defecation, or ambulation; (3) very poor body condition.

## Histology, TRAP staining, and toluidine blue staining of bone tissues

Following fixation in formalin for 2 days, the femurs, tibiae (with tumors), and calvariae of mice were decalcified in 12.5% EDTA solution for 5 days before being transferred to 70% ethanol. After paraffin embedding, the tissues were sectioned at 4 µm thickness. The slides

were deparaffinized in xylene, rehydrated in gradients of ethanol, and immersed in PBS for 5 min. For TRAP staining, we used a staining kit (Sigma, 387A-1KT) according to the manufacturer's instructions. Briefly, the slices were incubated with the staining buffer containing Fast Garnet GBC solution, Naphthol AS-BI phosphoric acid solution, acetate solution, and tartrate solution in a 37 °C water bath protected from light for 1 h. After being rinsed with water for 5 min, the slices were counterstained with methyl green (Vector Laboratories, H-3402–500) for 1 min and rinsed with water for 5 min. The slides were mounted with VectaMount Permanent Mounting Medium (Vector Laboratories, H-5000). For toluidine blue staining, the slides were stained with toluidine blue solution (Sigma, 89640), dehydrated and cleared with xylene, and coverslipped with DMX hydrophobic adhesive. All slides were scanned with an Aperio CS2 Digital Pathology Slide Scanner (Leica Biosystems). Bone histomorphometry was analyzed by using Bioquant OSTEO II software (Bioquant Nashville) on the sub-epiphyseal region 150 µm away from the distal growth plate and extending 1.3 mm into the bone compartment, at a distance of 150 µm from the cortical walls.

## Immunohistochemical staining

Sections of tibiae (with tumors) were deparaffinized in xylene and degraded alcohols. Heat-induced epitope retrieval was performed by using a 2100-Retriever. Slides were rinsed with PBS, and a hydrophobic barrier was created around the tissue using a hydrophobic barrier pen (Vector Laboratories, H-4000-2). Then, slides were placed in an incubating chamber with blocking solution (Vector Laboratories, SP-6000) for 10 min and rinsed with PBS, followed by incubation with 20% horse serum (Vector Laboratories, PK-7200) for 20 min. Next, slides were incubated with the primary antibody against RFP (1:400, Abcam, ab62341, RRID: AB_945213) at 4 °C overnight and rinsed with PBS, followed by incubation with a Horse Anti-Rabbit IgG Antibody (H + L), Biotinylated, R.T.U. (Vector Laboratories, BP-1100-50) or goat IgG HRP-conjugated antibody (R&D systems, HAF017) for 30 min. After being washed again with PBS, slides were incubated with the avidin-biotin detection complex (ABC; Vector Laboratories, SK-4100) for 30 min and were then developed with 3,3'-diaminobenzidine (DAB) solution (Vector Laboratories, SK-4100). Counterstaining was performed by using Hematoxylin QS (Vector Laboratories, H-3404). Slides were scanned with an Aperio CS2 Digital Pathology Slide Scanner (Leica Biosystems).

## Calcein staining of bone tissues

For dynamic histomorphometric measures of bone formation, calcein (Sigma, C0875) was intraperitoneally injected twice into mice (at 5 days and 1 day before euthanasia) at a dose of 25 mg/kg to obtain double labeling of newly formed bones. The non-decalcified femur bones were embedded in methyl methacrylate. The tissues were sectioned at 5 µm thickness, and the images were acquired by using an inverted microscope. Bone histomorphometric analysis of mineral apposition rate (MAR) was done with Bioquant OSTEO II software (Bioquant Nashville).

## Cell culture

The HEK293T (female) cell line was from Li Ma's lab stock (originally from the American Type Culture Collection, ATCC, CRL-3216). The L929 (male) cell line was from Dr. Dihua Yu (MD Anderson Cancer Center, Houston, TX). The RAW264.7 (male) and EO771 (female) cell lines were from Dr. Liuqing Yang (MD Anderson Cancer Center, Houston, TX). The U937 (male) cell line was from Dr. Xiang Zhang (Baylor College of Medicine, Houston, TX). The B16F1 (male) cell line was from MD Anderson's Cytogenetics and Cell Authentication Core. HEK293T, L929, RAW264.7, EO771, and B16F1 cell lines were cultured with Dulbecco's Modified Eagle Medium supplemented with 10% fetal bovine serum (FBS) and 1% penicillin/streptomycin. The U937 cell line

was cultured with RPMI 1640 supplemented with 10% FBS and 1% penicillin/streptomycin. Cells were maintained in a humidified, 5% $CO_2$ atmosphere at 37 °C, and low-passage stocks were maintained in a centralized lab cell bank. Short tandem repeat profiling and mycoplasma tests were done by ATCC and MD Anderson's Cytogenetics and Cell Authentication Core.

## Osteoclast differentiation

Osteoclast differentiation from bone marrow-derived macrophages (BMMs) was induced as described previously[71]. Briefly, femurs, tibiae, and iliac bones were removed from mice after euthanasia. Small incisions were made at both the proximal and distal ends of the bones, and the bones were placed in a sterile tube and centrifuged at $10,000 \times g$ at room temperature for 15 s. After purification with a 70 µm cell strainer, bone marrow cells were cultured in Minimum Essential Medium $\alpha$ (MEMα, Gibco, 41061029) containing 10% FBS for 1 day. Non-adherent cells were collected and seeded in 24- or 6-well plates and treated with 50 ng/mL of mouse M-CSF (Peprotech, 315-02) for 2 days, after which mouse soluble RANKL (Peprotech, 315-11) was added at a concentration of 100 ng/mL for additional culture for 4–6 days. Osteoclast differentiation from the RAW264.7 mouse cell line was induced as described previously[72]. Cells were seeded in 24-well or 6-well plates at a density of $2 \times 10^4$ or $5 \times 10^5$ cells per well. 50 ng/mL of mouse soluble RANKL (Peprotech, 315-11) was used to induce differentiation, and the culture medium was changed every 2 days. Osteoclasts derived from BMMs or RAW264.7 cells were identified as multinucleated (more than three nuclei) cells by TRAP staining (Sigma, 387A-1KT). Osteoclast differentiation from the U937 human cell line was induced as described previously[53,54]. Cells were seeded in 24-well or 6-well plates at a density of $5 \times 10^5$ or $3 \times 10^6$ cells per well. Cells were treated with 100 ng/mL of PMA for 2 days, followed by 50 ng/mL of human M-CSF (Peprotech, AF-300-25) and 100 ng/mL human soluble RANKL (Peprotech, AF-310-01) for 12–14 days. The culture medium was changed every 2 days. Osteoclasts derived from U937 cells were identified as TRAP+ cells. Osteoclast markers (NFATC1, CTSK, and TRAP5) were examined by qPCR and immunoblotting.

## Osteoblast differentiation

Osteoblast differentiation from bone marrow MSCs was induced as previously described[71]. Briefly, bone marrow cells were collected from the femurs, tibiae, and iliac bones of mice and plated for culture. After 48 h, non-adherent cells were removed, and attached cells were trypsinized and seeded in 12-well plates. When the cells reached 90% confluence, osteogenic differentiation medium (MEM containing 10% FBS, 5 mM β-glycerol phosphate, Selleck Chemicals, S3620, and 50 µg/ml of ascorbic acid, Selleck Chemicals, S3114) was added, and cells were cultured for 10–21 days. Then, ALP staining was done by using a staining kit (Sigma, 86R-1KT) according to the manufacturer's protocol. For ALP activity detection in the medium, equal volumes of conditioned medium were analyzed with an Alkaline Phosphatase Activity Fluorometric Assay Kit (BioVision, K422-500). For alizarin red S (ARS) staining, cells were fixed with 4% polyoxymethylene for 15 min and stained with 1% ARS (pH 4.2, Sigma-Aldrich, A5533) for 10 min. The dye was then removed and the cells were washed three times with water and photographed with an inverted microscope. The calcium mineralization stained with ARS was dissolved with 10% acetic acid and heated at 85 °C for 10 min, followed by neutralization with 10% ammonium hydroxide. The samples were transferred to 96-well plates, and the absorbance at 405 nm was measured on a microplate reader (Biotek Synergy 2).

## F-actin ring staining

RAW264.7 cells were treated with RANKL to induce differentiation into osteoclasts, fixed with 4% paraformaldehyde for 20 min, rinsed with PBS, and permeabilized with 0.5% Triton X-100 at room

temperature for 10 min. The cells were washed with PBS and blocked with 5% FBS at room temperature for 30 min. The fixed cells were stained with diluted phalloidin green 488 (1:100, BioLegend, 424201) in the dark for 20 min and mounted with the antifade mounting medium with DAPI (Vector Laboratories, H-1200-10). The slides were imaged with a Zeiss LSM880 confocal microscope and processed with Zen 2.6 (Zeiss) software.

## Plasmids

Mouse Nfatc1 was amplified from Prv-NFAT2 WT (Addgene, 11101) and Prv-NFAT2 CA (Addgene, 11102) by using PrimeSTAR Max DNA Polymerase (Takara, R045A). The resulting PCR products were subcloned into pDonor 201 through the Gateway BP reaction (Invitrogen, 11789020), and then cloned into the pBabe-SFB destination vector through the Gateway LR reaction (Invitrogen, 11-791-100). Mouse pDonor 223-Tead3 was obtained from the DNASU Plasmid Repository (https://dnasu.org/DNASU/Home.do) and cloned into pBabe-MYC and pLenti 6.2 FLAG-V5 vectors through the Gateway LR reaction (Invitrogen, 11-791-100). For cloning of truncation mutants, fragments of Nfatc1 or Tead3 were amplified and cloned into the pBabe-SFB or pBabe-MYC destination vector. Full-length mouse Malat1 and fragments (P1-P6) were amplified from pcDNA3.1-Malat1 (Li Ma's lab stock) and subcloned into PB-CAG-BGHpA (Addgene, 92161) by using an In-Fusion Cloning Kit (Takara, 638909). Primers used for cloning are listed in Supplementary Data 2.

## Lentiviral transduction

Lentiviruses were produced in HEK293T cells by co-transfection with the viral vector and packaging plasmids (pMD2.G: Addgene, 12259; psPAX2: Addgene, 12260). Two days after transfection, viral supernatant was harvested, filtered through a 0.45 µm filter, and added to target cells in the presence of polybrene reagent (Sigma, R-1003-G) at 4 µg/mL. The infected cells were selected with puromycin, hygromycin B, or blasticidin, as indicated below.

## Malat1 knockdown, knockout, and overexpression

To stably knockdown mouse Malat1, we designed gRNAs targeting mouse *Malat1* by using CHOPCHOP. The sgRNA sequences are listed in Supplementary Data 3. Primers were annealed and ligated into the pCRISPRia-v2 vector (Addgene, 84832) digested with BstXI and BlpI. RAW264.7 cells were infected with Lenti-dCas9-KRAB-blast (Addgene, 89567) lentivirus and selected with blasticidin (10 µg/mL). The surviving cells were then infected with pCRISPRia-v2-Malat1 lentivirus and selected with puromycin (10 µg/mL). Malat1 knockdown was verified by qPCR. To knockout MALAT1 in HEK293T cells, we infected the cells with lentiCas9-blast (Addgene, 52962) lentivirus and selected the cells with blasticidin (10 µg/mL). The surviving cells were infected with pDECKO_GFP (Addgene, 72619) or pDECKO_MALAT1_C (Addgene, 72622) lentivirus and selected with puromycin (1 µg/mL) as previously described[73]. After selection, single cells were plated in 96-well plates and cultured for 2 weeks. Malat1 knockout was verified by qPCR and DNA sequencing of individual clones. For restoration of Malat1 expression, MALAT1-knockout HEK293T cells were co-transfected with PB-CAG-BGHpA-Malat1 and Super PiggyBac Transposase (System Biosciences, PB210PA-1) by using polyethyleneimine hydrochloride MAX (Polysciences, 24765-1) and selected with hygromycin (300 µg/mL). For expressing full-length Malat1 or fragments (P1-P6) in MALAT1-knockdown U937 cells, PB-CAG-BGHpA-Malat1 or PB-CAG-BGHpA-Malat1 (P1-P6) was co-transfected with Super PiggyBac Transposase (System Biosciences, PB210PA-1) by using the 4D-Nucleofector system (Lonza Bioscience) according to the protocol of the SF Cell Line 4D-Nucleofector™ X Kit (Lonza Bioscience, V4XC-2012). Transfected U937 cells were selected with hygromycin (300 µg/mL). qPCR was used to verify the expression of full-length Malat1 and fragments.

## TEAD3 knockdown and overexpression

To overexpress mouse Tead3 in RAW264.7 cells, we used CRISPRa to activate endogenous Tead3 expression, considering that the translation of Tead3 is initiated at a non-AUG start codon[74,75]. The sgRNA sequences are listed in Supplementary Data 3. Primers were annealed and ligated into the Lenti sgRNA(MS2)_zeo backbone (Addgene, 61427) digested with BsmBI. RAW264.7 cells were infected sequentially with Lenti dCas9-VP64_blast (Addgene, 61425), and Lenti MS2-P65-HSF1_Hygro (Addgene, 61426), and were selected with blasticidin (10 µg/mL) and hygromycin (750 µg/mL). The surviving cells were then infected with Lenti sgRNA(MS2)_zeo-Tead3 lentivirus and selected with zeocin (1 mg/mL). Tead3 overexpression was verified by qPCR and immunoblotting. For TEAD3 knockdown in U937 cells, the following pLKO-puro constructs inserted with shRNA were purchased from Sigma: TRCN0000015948 (5′-GCCACTGTTCTGCGCTTTAAT-3′), and TRCN0000015949 (5′-CCATGTCTACAAGCTCGTCAA-3′). A non-targeting shRNA in the pLKO-puro backbone was used as the control. Infected U937 cells were selected with puromycin (3 µg/mL).

## RNA interference

The siRNA Universal Negative Control (Sigma, SIC001) and siRNAs targeting mouse *Tead3* or *Nfatc1* were synthesized by Sigma. For mouse *Tead3*, siRNA constructs SASI-Mm02-00301288 (5′-CGUCUA-CAAGCUUGUCAAAdTdT-3′) and SASI-Mm02-00301289 (5′-GCAA-GAUGUACGGUCGAAAdTdT-3′) were used. For mouse *Nfatc1*, siRNA constructs SASI-Mm02-00323571 (5′-CUCUCACGCUACAGCU-GUUUdTdT-3′) and SASI-Mm01-00029470 (5′-CCUCUGUGGCCCU-CAAAGUdTdT-3′) were used. siRNA oligonucleotides were transfected into RAW264.7 cells by using Lipofectamine RNAiMAX (Invitrogen, 13778075) according to the manufacturer's instructions. The knockdown efficiency was verified by immunoblotting.

## Cytoplasmic-nuclear fractionation

Control and Malat1-knockdown RAW264.7 cells were plated in 6-cm dishes. At 12 h after seeding, the cells were treated with soluble RANKL (50 ng/mL) for 3 days. Nuclear and cytoplasmic proteins were fractionated by using the NE-PER Nuclear and Cytoplasmic Extraction Kit (ThermoFisher Scientific, 78833) according to the manufacturer's protocol. After protein extraction, Western blot analysis was performed to detect Nfatc1 protein in the cytoplasmic and nuclear fractions. Gapdh and Lamin B1 were used as markers of the cytoplasm and the nucleus, respectively.

## Protein pulldown and immunoprecipitation

HEK293FT cells were transfected with SFB (a triple-epitope tag containing S-protein, FLAG, and streptavidin-binding peptide)-tagged Nfatc1 (full-length or truncation mutants) and MYC-tagged Tead3 (full-length or truncation mutants) and harvested 2 days after transfection. Cells were lysed in RIPA lysis buffer (Sigma, 20-188) at 4 °C for 15 min and sonicated. The lysates were centrifuged at 14,000 rpm at 4 °C for 15 min, and the supernatant was incubated with specific beads or antibodies. For the pulldown of SFB-tagged proteins, cell extracts were incubated with S-protein beads (EMD Millipore, 69704-3). For immunoprecipitation of MYC-tagged proteins, cell extracts were incubated with anti-MYC beads (Sigma, A7470, RRID: AB_10109522). After incubation at 4 °C overnight, the immune complexes were centrifuged and washed with PBS three times, and the bound proteins were eluted by boiling in Laemmli buffer at 95 °C for 10 min, followed by Western blot analysis with the indicated antibodies.

## Immunoblotting

Cells were lysed in RIPA lysis buffer (Sigma, 20-188, 10×: 0.5 M Tris-HCl, pH 7.4, 1.5 M NaCl, 2.5% deoxycholic acid, 10% NP-40, 10 mM EDTA) containing protease inhibitors and phosphatase inhibitors (GenDE-POT). Proteins were diluted in sample buffer (Bio-Rad), run on 4%-20% precast gradient gels (Bio-Rad), and transferred to a nitrocellulose membrane (Bio-Rad). After being blocked with 5% non-fat milk in Tris-buffered saline with 0.05% Tween-20, membranes were incubated with the primary antibody, followed by incubation with the anti-mouse (1:5000, Cytia, NXA931, RRID: AB_772209) or anti-rabbit (1:5000, Cytia, NA934, RRID: AB_772206) secondary antibody conjugated with horseradish peroxidase. After washing, the bands were detected with an enhanced chemiluminescent reagent (ThermoFisher Scientific, 34580). Primary antibodies used are as follows: antibodies against Nfatc1 (1:1000, Santa Cruz Biotechnology, sc-7294, RRID: AB_2152503), Tead1 (1:1000, Cell Signaling Technology, 12292 S, RRID: AB_2797873), Tead2 (1:1000, Proteintech, 21159-1-AP, RRID: AB_2861186), Tead3 (1:1000, Proteintech, 13120-1-AP, RRID: AB_2203068), Tead4 (1:1000, Proteintech, 12418-1-AP, RRID: AB_2203074), Ctsk (1:1000, Proteintech, 11239-1-AP, RRID: AB_2245581), Mitf (1:1000, Cell Signaling Technology, 12590 S, RRID: AB_2616024), Yap (1:1000, Cell Signaling Technology, 12395 S, RRID: AB_2797897), c-Fos (1:1000, Proteintech, 66590-1-Ig, RRID: AB_2881950), c-Jun (1:1000, Cell Signaling Technology, 9165 S, RRID: AB_2130165; 1:1000, Proteintech, 24909-1-AP, RRID: AB_2860574), phospho-c-Jun (S63) (1:1000, Cell Signaling Technology, 91952 S, RRID: AB_2893112), p65 (1:1000, Cell Signaling Technology, 8242 S, RRID: AB_10859369), phospho-p65 (S536) (1:1000, Cell Signaling Technology, 3033 S, RRID: AB_331284). Erk1/2 (1:1000, Cell Signaling Technology, 4695 S, RRID: AB_390779), phospho-Erk1/2 (Thr202/Tyr204) (1:1000, Cell Signaling Technology, 4370 S, RRID: AB_2315112), Jnk (1:1000, Cell Signaling Technology, 9252 S, RRID: AB_2250373), phospho-Jnk (Thr183/Tyr185) (1:1000, Cell Signaling Technology, 4668 S, RRID: AB_823588), IκBα (1:1000, Cell Signaling Technology, 9242 S, RRID: AB_331623), Creb1 (1:1000, Cell Signaling Technology, 9197 S, RRID: AB_331277; 1:1000, Proteintech, 12208-1-AP, RRID: AB_2245417), p38 (1:1000, Cell Signaling Technology, 8690 S, RRID: AB_10999090), Cas9 (1:1000, BioLegend, 844301, RRID: AB_2749904), Enterobacterio Phage MS2 Coat Protein (MCP) (1:1000, Sigma, ABE76-I, RRID: AB_2827507), FLAG tag (1:10000, Sigma, F7425, RRID: AB_439687, and F3165, RRID: AB_259529), MYC tag (1:2000, Cell Signaling Technology, 2278 S, RRID: AB_490778; 1:2000, Santa Cruz Biotechnology, sc-40, RRID: AB_627268), HA tag (1:2000, Cell Signaling Technology, 3724 S, RRID: AB_1549585; 1:5000, Santa Cruz Biotechnology, sc-7392, RRID: AB_627809), Hsp90 (1:2000, BD Biosciences, 610419, RRID: AB_397799), β-tubulin (1:2000, Proteintech, 10068-1-AP, RRID: AB_2303998), β-actin (1:4000, Santa Cruz Biotechnology, sc-47778, RRID: AB_626632), Gapdh (1:4000, Santa Cruz Biotechnology, sc-365062, RRID: AB_10847862), and Lamin B1 (1:1000, Cell Signaling Technology, 13435 S, RRID: AB_2737428).

## RNA extraction, cDNA synthesis, quantitative PCR (qPCR), and RT-PCR

Total RNA from cells was extracted by using a TRIzol reagent (Invitrogen, 15596026) or a PureLink RNA Mini Kit (Invitrogen, 12183018 A). cDNA was synthesized from 1 µg of total RNA by using an iScript cDNA Synthesis Kit (Bio-Rad, 1708891). Real-time PCR and data collection were done with SYBR Green Supermix (Bio-Rad, 1725124) on a CFX96 instrument (Bio-Rad). For cells, data were normalized to *ACTB*, *GAPDH*, or U6. For mouse tissues, data were normalized to 18 S rRNA. The primer sequences are listed in Supplementary Data 4. To verify the expression of Malat1 fragments (P1–P6) in MALAT1-depleted U937 cells, we ran the RT-PCR products on 3% agarose gels containing Gel-Green and detected the signals by using the ChemiDoc MP imaging system (Bio-Rad).

## Chromatin immunoprecipitation (ChIP) assay

ChIP assays of control and Malat1-knockdown RAW264.7 cells were done with a ChIP assay kit (Millipore, 17-371) as described previously[22]. Briefly, $1 \times 10^7$ RAW264.7 cells were cross-linked by using 1% formaldehyde. Excess formaldehyde was quenched by glycine, and cell

pellets were collected and lysed with SDS lysis buffer. The lysates were sonicated so that the chromosomal DNA fragments were 200–800 bp in length. Chromatin extracts were precleared with Protein G agarose, followed by immunoprecipitation with 8 µg of an Nfatc1-specific antibody (Invitrogen, MA3024, RRID: AB_2236037) or normal mouse IgG at 4 °C overnight. Immune complexes were collected on Protein G agarose beads and washed. The protein-DNA complexes were eluted with 1% SDS in 50 mM NaHCO3. After the reversal of protein-DNA cross-links and removal of proteins, purified DNA was used for PCR amplification of the *Ctsk*, *Nfatc1*, and *Acp5* promoter regions bound to Nfatc1. The *Ctsk*, *Nfatc1*, and *Acp5* gene-specific primers were described previously[52,76,77] and are listed in Supplementary Data 5.

## RNA pulldown assay

Full-length mouse Malat1 (NR_002847) was divided into six non-overlapping pieces (P1-P6, 1.1–1.2 kb each) and cloned into the pGEM-T vector as previously described[22]. The vector was linearized by NotI-HF (New England Biolabs, R3189s) and used as the template for the synthesis of biotin-labeled RNA by using an in vitro T7 transcription kit (New England Biolabs, E2040S). Biotin-16-UTP (Roche, 11388908910) was used to biotinylate the RNAs. Non-biotinylated RNAs and biotinylated U1 were synthesized as negative controls. After in vitro transcription, the products were purified by using a PureLink RNA Mini Kit (Invitrogen, 12183018 A) and digested with RNase-free DNase I (Invitrogen, 12185010) to remove the template DNA according to the manufacturer's protocols. 3 µg of purified biotin-labeled or biotin-free RNA was heated at 90 °C for 2 min and chilled on ice for 2 min in the RNA structure buffer (2×: 20 mM Tris-HCl at PH 7.4, 0.2 M KCl, 20 mM MgCl₂, 2 mM DTT) containing RNase inhibitor (Takara, 2313B). RNA samples were placed at room temperature for 20 min for proper secondary structure formation. HEK293T cells overexpressing V5-tagged Tead3 were lysed in RIPA lysis buffer (Sigma, 20-188) containing protease inhibitors (GenDEPOT) and RNase inhibitor and sonicated. Cell lysates were precleared with streptavidin agarose (Pierce, 20349) at room temperature. 3 mg of precleared cell lysates were added to each folded RNA sample and incubated at room temperature overnight. Streptavidin agarose was added and incubated at room temperature for 1 h. The bound proteins were washed and eluted by boiling in Laemmli buffer at 95 °C for 10 min and subjected to Western blot analysis.

## RIP assay

The RIP assay was done with an EZ-Magna RIP Kit (Millipore, 17-10522) according to the manufacturer's instructions. Briefly, ~4 × 10⁷ RAW264.7 cells were lysed in the Complete Nuclei Isolation Buffer (included in the kit) with protease inhibitor cocktail and RNase inhibitor and centrifuged at 800 × g at 4 °C for 5 min. The nuclear pellet was resuspended with Complete RIP Lysis Buffer (included in the kit). After being treated with DNase I, the samples were precleared with Protein A/G magnetic beads and incubated with Protein A/G magnetic beads coated with a pan-Tead-specific antibody (Cell Signaling Technology, 13295 S, RRID: AB_2687902), a Tead3-specific antibody (Proteintech, 13120-1-AP, RRID: AB_2203068), or normal Rabbit IgG at 4 °C overnight. The beads were placed on a magnetic separator and washed with Nuclear RIP Wash Buffer, and the bead-bound immunoprecipitates were subjected to RNA purification by using a PureLink RNA Mini Kit (Invitrogen, 12183018 A), followed by DNase I treatment. Purified RNA samples were used for cDNA synthesis, followed by qPCR analysis with the primers listed in Supplementary Data 4. U6 was used as a negative control. The results are presented as fold enrichment (normalized to IgG).

## Luciferase reporter assay

The pGL-NFAT reporter construct containing 3× NFAT-binding sites was from Addgene (17870). The mouse *Ctsk* promoter region was PCR-amplified (PCR primers are listed in Supplementary Data 2) and ligated into the linearized pGL3-basic plasmid by using an In-Fusion HD Cloning Kit (Takara Bio, 638909). MALAT1 wild-type, MALAT1-knockout, and Malat1-restored HEK293T cells were plated in triplicates in 96-well plates. The next day, 16.67 ng of the indicated firefly luciferase vector, 50 ng of Nfatc1-CA-SFB, 1 ng of a Renilla luciferase vector, and 20 or 100 ng of Tead3-MYC were transfected per well. At 36 h after transfection, firefly and Renilla luciferase activities were measured by using a Dual-Luciferase Reporter Assay (Promega, E1910) on a microplate reader according to the manufacturer's protocol. Firefly luciferase activity was normalized to Renilla luciferase activity.

## ELISA

Mouse blood was collected through intracardiac puncture immediately after euthanasia. The blood was transferred to a 1.5 mL tube and left at room temperature for 30 min. The clot was removed by centrifuging at 7000 rpm in a refrigerated centrifuge for 15 min. The resulting supernatant was aliquoted and stored in a −80 °C freezer. For the detection of serum TRAP5b, a TRAP5b ELISA Kit (Immunodiagnostic Systems, SB-TR103) was used according to the manufacturer's instructions.

## Bulk RNA-seq analysis

The RNA-seq data of HSCs and different hematopoietic lineages from umbilical cord blood of healthy individuals (E-MTAB-3819) or mouse bone marrow (E-MTAB-7391) were downloaded from Expression Atlas (https://www.ebi.ac.uk/gxa/experiments). A heatmap of differentially expressed genes (DEGs) was generated by using R language (Version 4.3.2). Human MALAT1 expression was compared among HSCs, multipotent progenitors (MPPs), and CMPs. The expression values in each group are from three or four healthy individuals with 1–4 replicates. Mouse Malat1 expression was compared among HSCs and four hematopoietic multipotent progenitors (MPPs): MPP1, lineage-Sca1+-Kit+CD135+CD150-CD48+CD34+; MPP2, lineage-Sca1+Kit+CD135-CD150+CD48+CD34+; MPP3, lineage-Sca1+Kit+CD135-CD150+CD48-CD34+; and MPP4, lineage-Sca1+Kit+CD135-CD150-CD48+ CD34+. The expression values in each group are from four samples, and each sample contains cells pooled from three mice.

The RNA-seq data (GSE38747) of CD14+ macrophages and the derived MGCs were downloaded from Gene Expression Omnibus (GEO; http://www.ncbi.nlm.nih.gov/geo/). After the probe ID was converted to the gene symbol based on the annotation of the platform, the data were normalized, and an expression matrix was obtained. The DEGs were identified by limma package in R with the following cutoff values: |log₂ (fold change)| > 1 and *P* value < 0.001. The heatmap and volcano map of the DEGs were created by R. GSEA of the RNA-seq data was performed by using clusterProfiler and enrichplot package in R[78]. Gene sets with an adjusted *P* value of less than 0.05 were considered significantly enriched and are listed in Supplementary Data 1.

## Single-cell RNA-seq (scRNA-seq) analysis

scRNA-seq analysis was performed on publicly available datasets. The following data were downloaded from the Gene Expression Omnibus (https://www.ncbi.nlm.nih.gov/geo/): GSE190772 with samples from two patients with breast cancer bone metastases[38,43], GSE162454 with samples from six osteosarcoma patients[44,45], and GSE169396 featuring bone tissues from a non-osteoporotic individual and three osteoporosis patients[46].

Standard procedures were used to process the scRNA-seq data. The count matrix was read by using the Read10X function of the Seurat package (Version 4.3.0) and further converted to the dgCMatrix format. We filtered out low-quality cells by using the following criteria: genes expressed in less than three cells were deleted, and cells expressing <200 genes were deleted. The merge function was used to integrate all individual objects into an aggregate object, and the

RenameCells function was used to ensure unique cell labels. A global-scaling normalization method ("LogNormalize") was used to ensure equal total gene expression in each cell, with the scale factor set to 10,000. The top 2000 variably expressed genes were used in downstream analysis by using the FindVariableFeatures function. The ScaleData function, "vars.to.regress" option UMI, and percent mitochondrial content were used to regress out unwanted sources of variation. Principal component analysis incorporating highly variable features was used to reduce the dimensionality of the dataset, and the first 30 PCs were identified for analysis. The "Harmony" method[47] was used to remove batch effects between samples. The scrublet algorithm was used to remove potential doublets. Cell clustering was performed based on the edge weights between any two cells, and a shared nearest-neighbor graph was produced by using the Louvain algorithm, which is implanted in the FindNeighbors and FindClusters functions. The FindClusters function's resolution parameter was used repeatedly between 0.05 and 1. The clustree function was used to observe cell clustering trees at different resolutions.

Cell annotation was performed by using multiple methods. First, the Immune_All_High atlas in CellTypist software was used for automatic annotation. Then, the FindAllMarkers function in the Seurat package was used to identify markers for each cell type, and these cell markers were checked in the Cell Taxonomy database (https://ngdc. cncb.ac.cn/celltaxonomy/). The necessary manual adjustment was made to ensure reliable annotation results.

### Statistical analysis

Except for the animal studies, each experiment was repeated at least three times with similar results. Statistical analyses were done with GraphPad Prism 9.0.0. Unless otherwise noted, data are presented as mean ± s.e.m., and a two-tailed unpaired $t$-test was used to compare two groups of independent samples. Statistical methods used for single-cell and bulk RNA-seq analyses and Expression Atlas data analysis are described above. $P < 0.05$ was considered statistically significant.

### Reporting summary

Further information on research design is available in the Nature Portfolio Reporting Summary linked to this article.

## Data availability

All data supporting the findings of this study are available within the Article, Supplementary Information, or Source Data file. Previously published data used in this study are E-MTAB-3819: https://www.ebi.ac.uk/biostudies/arrayexpress/studies/E-MTAB-3819; E-MTAB-7391: https://www.ebi.ac.uk/biostudies/arrayexpress/studies/E-MTAB-7391; GSE38747: https://www.ncbi.nlm.nih.gov/geo/query/acc.cgi?acc=GSE38747; GSE190772: https://www.ncbi.nlm.nih.gov/geo/query/acc.cgi?acc=GSE190772; GSE162454: https://www.ncbi.nlm.nih.gov/geo/query/acc.cgi?acc=GSE162454; and GSE169396: https://www.ncbi.nlm.nih.gov/geo/query/acc.cgi?acc=GSE169396. Source data are provided with this paper.

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

## Acknowledgements
We thank MD Anderson's Small Animal Imaging Facility, Bone Histo-morphometry Core, Flow Cytometry and Cellular Imaging Core, Functional Genomics Core, Advanced Technology Genome Core, and Cytogenetics and Cell Authentication Core for technical assistance. We are grateful to all members of the Ma Lab for the discussion and to Christine Wogan for the critical reading of the manuscript. L.M. is supported by US National Institutes of Health (NIH) grants R01CA166051 and R01CA269140, an American Cancer Society grant (award number: DBG-22-161-01-MM), and the Nylene Eckles Distinguished Professorship of MD Anderson Cancer Center. The core facilities are supported by MD Anderson's Cancer Center Support Grant P30CA016672 from the NIH.

## Author contributions
Y.Z., Y.S. and L.M. conceived and designed the study. Y.Z. performed most experiments and data analyses. J.N. performed single-cell RNA-seq analysis and other bioinformatic analyses. H.T., Y.D., M.S., C.M., J.Z., A.T., and F.Y. performed some experiments and provided technical assistance. L.S. and H.W. reviewed the data and provided significant intellectual input. Y.S. generated some reagents and contributed protocols. S.N. provided the Malat1-knockout mice. Y.Z. and L.M. wrote the manuscript with input from all other authors. L.M. provided scientific direction, established collaborations, and allocated funding for this study.

## Competing interests
The authors declare no competing interests.
