## [Peer Review File · Nature Communications]

Long noncoding RNA Malat1 protects against osteoporosis and bone metastasisREVIEWER COMMENTS

Reviewer #2 (Remarks to the Author):

Major comments

(1) The observation that Tead3-binding sites may be distributed diffusely on Malat1 is interesting. However, it makes it challenging to elucidate and test how exactly Tead3/Malat1 is functioning. Further corroboration, specifically manipulating Tead3/Malat1 binding to fully establish whether this is noisy, non-functional binding or a necessity for function.

(2) In addition to the RNA pulldown assays, a strategy to assess Tead3 interactions with RNA should be evaluated.

(3) Further confirmation in an additional endogenous setting showing the interaction between Tead3/Malat1 would strengthen the applicability of the proposed model.

(4) While there in vivo work shows the potential contribution of MALAT1 to bone metastasis, the study does not actually confirm their proposed regulatory model occurs in vivo. This is critical for supporting the relevance of their model.

(5) More information showing that the proposed regulatory model occurs in human patients is important to understand whether these findings could actually translate into human cancer.

(6) The incorporated of single cell data would help support the role of each gene in their proposed models within bone metastasis.

Minor

(GSEA in 1d would help to use all genes, not just differentially expressed

Reviewer #3 (Remarks to the Author):

NCOMMS-23-02783-T

Long noncoding RNA Malat1 inhibits Tead3-Nfatc1-mediated osteoclastogenesis to suppress osteoporosis and bone metastasis

This work by Zhao et al. investigated a role of MALAT1, one of the few highly conserved lncRNAs in osteoclast differentiation. Using conventional mouse osteoclastic culture derived, mice models of osteoporosis and bone metastasis, they demonstrated that MALAT1 plays an essential role in osteoclasts and bone pathophysiology. This was also possibly true in humans through evidences from their in vivo

human osteoclast culture experiments and GWAS search. Furthermore, their molecular interaction analyses in vitro and in silico demonstrated that Malat1 binds to Tead3; most predominantly expressed in osteoclasts among Tead family member, thus sequestering Tead3 from its binding to Nfatc1. The conclusion of this work appeared to be well supported by their substantial amount of experimental evidences, therefore implying a relevant message in human bone pathophysiology and bone biology. Some concerns could be seen in the role of TEAD3 in osteoclastogenesis. My specific comments are below.

Specific comments;

Knockdowns of Tead3 in RAW264.7 cells reduced the RANKL-induced osteoclast differentiation accompanied with the downregulated expression of Nfatc1 and Ctsk (Extended Data Fig. 7). However, the data shown in Fig.4q demonstrated that Knockdowns of Tead3 in the same cells depleting MALTA1 showed comparable scores of number of osteoclasts/well with the control. These experiments did not appear to be consistent, and not entirely support the mechanistic interpretation shown in Fig.4s. Further evidence for the role of TEAD3 in osteoclast differentiation and careful interpretation would be required.

Reviewer #4 (Remarks to the Author):

In this manuscript, the authors study the function and mechanism of lncRNA Malat1 in promoting osteoporosis and bone metastasis in in vitro and in vivo models. They found that Malat1 was downregulated during the osteoclast differentiation. A Murine model of Malat1 knockout promoted osteoporosis, that was rescued by Malat1 restoration. Multiple pull-down and IP assays indicated that Tead3 and Nfatc1 are critical in mediating these effects. Although the experiments are conducted well with proper controls and data support the hypotheses in general, there are areas of concern that need to be addressed as listed below.

1. Based on the publicly available high-throughput sequencing datasets (extended data Fig. 1a) and gene expression (Fig. 1a), there are other genes that have the similar expression levels compared to Malat1. Why did the authors specifically select Malat1 as top-ranking candidate instead of other ones?
2. It's curious why the authors select sgRNA-2 and sgRNA-3 from B16F1 cell line and use them in RAW264.7 cells. Also, the knockout efficiency is not great, especially for sgRNA-3. Have the authors screened the knockout efficiency of different sgRNAs in RAW264.7 cells, or tried other genetic manipulation methods?
3. In Supplementary Fig. 1 and corresponding Methods section, please change the average expression value to individual expression values, since this is a very small cohort (are there any larger cohorts that can be included in this study?) and any outlier value would potentially affect the average value.
4. In Fig. 1e-g, in addition to RAW264.7, have the authors tried to use human macrophage/monocyte cell line to examine the role of Malat1 during osteoclastogenesis?
5. In Fig. 2, have the authors confirmed the expression of Malat1 in different organs of the genetic mouse models?

6. In Fig. 2o, the authors used one B16F1 melanoma cell intratibial bone metastasis model. Intratibial injection is not an ideal method to study bone metastasis, even though I understand the other methods such as intracardiac or intra-caudal arterial may present some technical issues. Since bone metastasis is one of the major conclusions, this should be more carefully addressed. Maybe the authors can try different methods or multiple models using the same method. Otherwise, I'm not confident this data is sufficient to support the conclusion that Malat1 deficiency promotes bone metastasis.
7. The genetic model authors used are appropriate to test their hypothesis. However, the rescue phenotypes are not exactly same in male mice and female mice (Extended Fig. 2a, b, c, d). The authors claim that Malat1 has functional relationship with AR in prostate cancer and generally women are more at risk of developing osteoporosis than men. Have the authors examined a role for AR along with Malat1 in osteoporosis?
8. Fig. 3g shows outliers in two knockdown groups. If the outliers were excluded, the average would not be significantly different than controls.
9. The suggestion that the nuclear accumulation of Nfatc1 in Extended Data Fig. 4d is not convincing since the difference is subtle and perhaps not run on the same gel? Total lysate bands should be compared on the same gel.
10. In Fig. 4b, it seems that Tead3 may be selectively expressed in BMMs, but RAW264.7 displays Tead-2, 3, 4 expression. And L929 seems to have a higher Tead3 expression. Please correct corresponding statements in the text.
11. There are several other proteins that also interact with Nfatc1 (Extended data Fig. 5a). The authors should test other candidates as well as Tead3 in their experiments to show that the effects are specific to Nfatc1 and Tead3 interaction.
12. The ChIP-qPCR verified the occupancy at the Nfatc1 regions. However, the results would be more significant if they carry out ChIP-seq at the genome-wide level instead of a single target.
13. Based on the hypothesis and model (Fig. 4s), Malat1 acts a decoy suppressor protein. Maybe the authors can try to overexpress Malat1 and treat with RANKL and compare the downstream targets with WT. If overexpression of Malat1 suppresses osteoporosis compared to WT that would further support their hypothesis.
14. Is there any clinical data that would support the role of Malat1 in osteoporosis? Do patients who have osteoporosis display low expression levels of Malat1?

Minor:

1. Some main figures contain too many sub-figures and appear too crowded. The authors should consider moving some of them to extended figures.

Point-by-point rebuttal to the reviewers
Nature Communications manuscript NCOMMS-23-02783-T

We thank Nature Communications for the interest in our study and the invitation to resubmit this manuscript. We are delighted the reviewers found that “the conclusion of this work appeared to be well supported by their substantial amount of experimental evidences, therefore implying a relevant message in human bone pathophysiology and bone biology”, and that “the experiments are conducted well with proper controls and data support the hypotheses in general”. We thank all reviewers for their constructive comments on how to improve this manuscript. After an 8-month revision, we thoroughly and attentively addressed the critiques with extensive new data (which expanded 4 main figures to 8 main figures, plus 10 supplementary figures) and further clarification. Revisions in the manuscript text are marked in red color.

Reviewer #1: No points to be addressed.

Reviewer #2:
Major comments

(1) The observation that Tead3-binding sites may be distributed diffusely on Malat1 is interesting. However, it makes it challenging to elucidate and test how exactly Tead3/Malat1 is functioning. Further corroboration, specifically manipulating Tead3/Malat1 binding to fully establish whether this is noisy, non-functional binding or a necessity for function.

Re: We agree with this reviewer that it’s challenging to assess whether the Malat1-Tead3 interaction is functional. Tead(3) is an unconventional RNA-binding protein and lacks the consensus binding sequence on RNA. Instead, both our previous paper¹ and the present manuscript presented evidence that the Tead(3)-binding sites are distributed diffusely on Malat1. Thus, it is not feasible to make Tead3 binding-deficient mutants of Malat1.

Nevertheless, we attempted to address this point by depleting Malat1 and adding back full-length Malat1 or Malat1 fragments (P1-P6). However, we failed to generate Malat1-knockout RAW264.7 cells by using CRISPR-based methods after we analyzed >200 clones, likely because RAW264.7 cells are not easy to transfect².

We then switched to the U937 cell line, a human monocyte/pre-osteoclast cell line. By using the CRISPRi method, we were able to deplete MALAT1 by >99% (**new data, Supplementary Fig. 7c**). Functional assays showed that MALAT1 depletion in U937 cells promoted osteoclast differentiation (**new data, Supplementary Fig. 7d, e**). Next, we performed osteoclast differentiation assays by treating the control, MALAT1-depleted, and Malat1-restored (with full-length Malat1 or Malat1 fragments, P1-P6) U937 cells with phorbol 12-myristate 13-acetate (PMA) (100 ng/mL) for 2 days, followed by 12-14 days of human M-CSF (50 ng/mL) and human soluble RANKL (100 ng/mL) treatment^{3, 4}. These assays showed that ectopic expression of each of the six Malat1 fragments in MALAT1-depleted U937 cells partially reversed, while re-expression of full-length Malat1 completely reversed RANKL-induced osteoclastogenesis (**new data, Supplementary Fig. 8b-d**).

(2) In addition to the RNA pulldown assays, a strategy to assess Tead3 interactions with RNA should be evaluated.

(3) Further confirmation in an additional endogenous setting showing the interaction between Tead3/Malat1 would strengthen the applicability of the proposed model.

Re: As requested in points #2 and #3, we performed RNA immunoprecipitation (RIP) assays with RAW264.7 cells by using a Tead3-specific antibody, showing that endogenous Malat1 was enriched in endogenous Tead3 immunoprecipitates relative to the control IgG group (**new data, Fig. 6c**).

(4) While there *in vivo* work shows the potential contribution of MALAT1 to bone metastasis, the study does not actually confirm their proposed regulatory model occurs *in vivo*. This is critical for supporting the relevance of their model.

Re: In the revised manuscript, we have added extensive new data to corroborate that Malat1 protects against osteoporosis (Figure 2) and bone metastasis (of both melanoma and mammary tumor cells; **new data, Figure 3**) – also supported by single-cell RNA-seq data from patients with osteoporosis or bone metastasis (**new data, Figure 4**), and that Malat1 binds Tead3 to inhibit Nfatc1 activity and osteoclastogenesis (**revised Figures 5-7**). While Nfatc1 is a well-established master regulator of osteoclastogenesis, we agree with this reviewer that we have not validated the mechanistic model *in vivo*. Proof of this point would require genetic rescue experiments (i.e., the rescue of the phenotypes of Malat1 knockout mice) by using osteoclast-specific Nfatc1 and Tead3 knockout mouse models, which will take several more years. In light of this constraint, we changed the title of this paper to “Long noncoding RNA Malat1 protects against osteoporosis and bone metastasis”, which represents a very important discovery and is worth reporting.

(5) More information showing that the proposed regulatory model occurs in human patients is important to understand whether these findings could actually translate into human cancer.

(6) The incorporation of single-cell data would help support the role of each gene in their proposed models within bone metastasis.

Re: We fully agree with points #5 and #6 that it would be beneficial to evaluate the human disease relevance of our findings at the single-cell level. To assess the clinical relevance of MALAT1 in osteoporosis and bone metastasis, we analyzed single-cell RNA-seq data from human bone tissues. The datasets included GSE190772 with samples from two patients with breast cancer bone metastasis^{5, 6}, GSE162454 with samples from six osteosarcoma patients^{7, 8}, and GSE169396 featuring bone tissues from a non-osteoporotic individual and three osteoporosis patients (femoral head collected during hip replacement surgery)⁹. Osteosarcomas and breast cancer bone metastases often exhibit osteolytic features.

We used the “Harmony” method¹⁰ to remove batch effects between samples, subsequently applying dimensionality reduction to annotate cell types based on marker genes (**new data, Supplementary Fig. 4a-c**). These analyses defined the cell cluster-specific transcriptome of different patient groups. We then analyzed the expression of MALAT1 in pre-osteoclasts (including monocytes and macrophages) and mature osteoclasts of the non-osteoporotic individual (**new data, Fig. 4a, b**), osteoporosis patients (**new data, Fig. 4c, d**), osteosarcoma patients (**new data, Fig. 4e, f**), and patients with breast cancer bone metastases (**new data, Fig. 4g, h**). Within each group, MALAT1 expression levels were significantly lower in osteoclasts compared with pre-osteoclasts (**new data, Fig. 4b, d, f, and h**). Moreover, across the four patient groups, MALAT1 expression levels in pre-osteoclasts and osteoclasts were significantly lower in patients with osteoporosis, osteosarcoma, or breast cancer bone metastasis than in the non-osteoporotic individual (**new data, Fig. 4i-k**). These findings indicate that reduced MALAT1 expression in the osteoclast lineage is associated with osteoporosis and bone lesions, including breast cancer metastases and osteosarcomas.

Minor

GSEA in 1d would help to use all genes, not just differentially expressed.

Re: As requested, we re-analyzed the data by using all genes (**revised Fig. 1d and revised Supplementary Table 1**). The new GSEA data are consistent with the previous analysis.

Reviewer #3:

This work by Zhao et al. investigated a role of MALAT1, one of the few highly conserved lncRNAs in osteoclast differentiation. Using conventional mouse osteoclastic culture derived, mice models of osteoporosis and bone

metastasis, they demonstrated that MALAT1 plays an essential role in osteoclasts and bone pathophysiology. This was also possibly true in humans through evidences from their in vivo human osteoclast culture experiments and GWAS search. Furthermore, their molecular interaction analyses in vitro and in silico demonstrated that Malat1 binds to Tead3; most predominantly expressed in osteoclasts among Tead family members, thus sequestering Tead3 from its binding to Nfatc1. **The conclusion of this work appeared to be well supported by their substantial amount of experimental evidences, therefore implying a relevant message in human bone pathophysiology and bone biology.** Some concerns could be seen in the role of TEAD3 in osteoclastogenesis. My specific comments are below.

Specific comments:

Knockdowns of Tead3 in RAW264.7 cells reduced the RANKL-induced osteoclast differentiation accompanied with the downregulated expression of Nfatc1 and Ctsk (Extended Data Fig. 7). However, the data shown in Fig.4q demonstrated that Knockdowns of Tead3 in the same cells depleting MALAT1 showed comparable scores of number of osteoclasts/well with the control. These experiments did not appear to be consistent, and not entirely support the mechanistic interpretation shown in Fig.4s. Further evidence for the role of TEAD3 in osteoclast differentiation and careful interpretation would be required.

Re: We are very grateful to this reviewer for appreciating the importance of this work in bone biology and pathophysiology. Regarding the experiments in Extended Data Fig. 7 and Fig. 4q of the initially submitted manuscript, they were performed at different times, and thus we would not compare the absolute numbers of osteoclasts per well from two different experiments. Nevertheless, We agree that the role of TEAD3 in osteoclastogenesis needs careful interpretation and further evidence.

It should be noted that Tead3 is upregulated during osteoclast differentiation (Fig. 6e, f). In Extended Data Fig. 7 of the initially submitted manuscript (now Fig. 7i-l), knockdown of Tead3 in RAW264.7 cells downregulated Nfatc1 and Ctsk expression and attenuated osteoclast differentiation, suggesting that upregulation of Tead3 has a functional role in osteoclastogenesis. In original Fig. 4q (now Fig. 7o), knockdown of Tead3 in Malat1-depleted RAW264.7 cells showed comparable numbers of osteoclasts with the control RAW264.7 cells, i.e., knockdown of Tead3 in Malat1-knockdown RAW264.7 cells reversed Malat1 depletion-induced osteoclast differentiation. These experiments suggest that Tead3 mediates osteoclast differentiation induced by Malat1 loss. This is consistent with our model (Figure 8) in which Malat1 binds and sequesters Tead3, blocking Tead3 from associating with Nfatc1 and inducing the transcription of Nfatc1 target genes, including *Nfatc1* itself and *Ctsk*. In response to RANKL stimulation, downregulation of Malat1 releases Tead3, thereby enhancing both the Tead3-Nfatc1 interaction as well as the transcription factor occupancy of Nfatc1 target genes, which leads to activation of Nfatc1-mediated gene transcription and osteoclast differentiation.

As requested, we performed additional experiments to provide further evidence for the role of TEAD3 in osteoclast differentiation:

1) To overexpress Tead3 in RAW264.7 cells, we used CRISPR activation (CRISPRa) to activate endogenous Tead3 expression (**new data, Fig. 7a, b**), considering that the translation of Tead3 is initiated at a non-AUG start codon^{11, 12}. We found that overexpression of Tead3 promoted osteoclast differentiation (**new data, Fig. 7c, d**) and upregulated Nfatc1, Trap5, and Ctsk expression (**new data, Fig. 7e-h**). Taken together with the loss-of-function results, these data corroborated that Tead3 is a positive regulator of osteoclast differentiation.

2) Similar to RAW264.7 cells (Fig. 6e), during osteoclast differentiation of the human pre-osteoclast cell line U937, TEAD3 was also the most upregulated TEAD family member (**new data, Fig. 6f**). Thus, we examined TEAD3's function in U937 cells, finding that shRNA-mediated knockdown of TEAD3 impaired human osteoclastogenesis (**new data, Supplementary Fig. 10a-c**) and downregulated the expression of NFATC1 and TRAP at both mRNA and protein levels (**new data, Supplementary Fig. 10d-f**). It should be noted that MALAT1 depletion did not affect the upregulation of TEAD3 during osteoclast differentiation (**new data, Supplementary**

Fig. 10g), suggesting that MALAT1 does not regulate TEAD3's expression levels (but instead inhibits the TEAD3-NFATC1 interaction).

Through the above experiments, we demonstrated that Tead3 promotes osteoclast differentiation in both human and mouse pre-osteoclasts. Tead3 functions by binding to Nfatc1 to promote Nfatc1's transcriptional activity; this is negatively regulated by Malat1, which binds and sequesters Tead3 to block its inaction with Nfatc1.

Reviewer #4:

In this manuscript, the authors study the function and mechanism of lncRNA Malat1 in promoting osteoporosis and bone metastasis in in vitro and in vivo models. They found that Malat1 was downregulated during the osteoclast differentiation. A Murine model of Malat1 knockout promoted osteoporosis, that was rescued by Malat1 restoration. Multiple pull-down and IP assays indicated that Tead3 and Nfatc1 are critical in mediating these effects. Although the experiments are conducted well with proper controls and data support the hypotheses in general, there are areas of concern that need to be addressed as listed below.

Re: We thank this reviewer for recognizing that “the experiments are conducted well with proper controls and data support the hypotheses in general”, and for the insightful points that we address as follows:

1. Based on the publicly available high-throughput sequencing datasets (extended data Fig. 1a) and gene expression (Fig. 1a), there are other genes that have the similar expression levels compared to Malat1. Why did the authors specifically select Malat1 as top-ranking candidate instead of other ones?

Re: This study focused on Malat1, because: Previously, through targeted inactivation, genetic rescue, and transgenic overexpression in mouse models, our lab discovered that the lncRNA MALAT1 sequesters TEAD proteins to suppress breast cancer lung metastasis¹. LncRNAs usually have low evolutionary conservation, whereas MALAT1 is one of the few highly conserved lncRNAs with abundant expression in normal tissues. It's unclear why normal tissues express such a highly abundant and conserved lncRNA.

Recent genome-wide association studies (GWAS) showed that single nucleotide polymorphisms (SNPs) are associated with osteoporosis^{13, 14}. Interestingly, one such analysis identified an SNP at the *MALAT1* locus that was associated with low bone mineral density (BMD)¹⁵. However, functional evidence of MALAT1 alterations having a role in low BMD and osteoporosis is lacking. On a quest to find the physiological roles of this highly abundant and conserved lncRNA, we found that MALAT1 is downregulated during osteoclast differentiation in humans and mice (Figure 1), which prompted us to study the function of MALAT1 in bone homeostasis and metastasis by using cell models and our genetically engineered mouse models, leading to our discovery that MALAT1 protects against osteoporosis (Figure 2) and bone metastasis (Figure 3).

In Fig. 1a, b, we analyzed gene expression during the differentiation of human placental macrophages into multinucleated giant cells (MGCs), in which osteoclasts are the major cell population¹⁶. Compared with macrophages, MGCs showed upregulation of known osteoclast markers and downregulation of MALAT1. As this reviewer pointed out, this RNA-seq analysis also identified other genes with significant expression changes during osteoclastogenesis. Thus, we intend to use MALAT1 as an entry point to elucidate the molecular and cellular events underlying bone homeostasis and metastasis.

In future work, we will perform gene set enrichment analysis and pathway analysis of genes that are downregulated or upregulated in MGCs. Genes associated with skeletal morphogenesis or homeostasis pathways will be of particular interest. Moreover, we will analyze single-cell RNA-seq data from patients with osteoporosis or bone metastasis to determine whether any of the differentially expressed genes (besides MALAT1; see our response to point #14 below) exhibit substantial differences between pre-osteoclasts and osteoclasts, and whether among different groups, their expression levels in pre-osteoclasts and osteoclasts are significantly different between non-osteoporotic donors and patients with osteoporosis or breast cancer bone metastasis. Then, genes

identified from these analyses will be subjected to mouse modeling for in-depth dissection of their functions in osteoporosis and bone metastasis of breast cancer and other cancer types that often exhibit osteolytic features.

2. It's curious why the authors select sgRNA-2 and sgRNA-3 from B16F1 cell line and use them in RAW264.7 cells. Also, the knockout efficiency is not great, especially for sgRNA-3. Have the authors screened the knockout efficiency of different sgRNAs in RAW264.7 cells, or tried other genetic manipulation methods?

Re: We designed 11 sgRNAs targeting mouse Malat1 for CRISPRi and tested their efficiency by using the B16F1 mouse melanoma cell line, because the B16F1 cell line is easy to transfect, whereas the RAW264.7 cell line is known to be hard to transfect². We found that sg2 and sg3 had higher efficiency than other sgRNAs (Supplementary Fig. 5a), and thus we used sg2 and sg3 in RAW264.7 cells, which achieved >50% knockdown efficiency (Fig. 5e). This is comparable to previous studies that used CRISPRi to knock down MALAT1 in human cancer cell lines, achieving a knockdown efficiency of 40-60%^{17,18}.

Malat1 is a nuclear RNA. Many previous studies used RNAi or ASOs for loss-of-function studies of Malat1. However, using RNAi for nuclear lncRNAs is questionable¹⁹; moreover, substantial off-target effects have been reported for ASOs²⁰⁻²². In this study, we attempted to generate Malat1-knockout RAW264.7 cells by using CRISPR-based methods; however, this effort failed after we analyzed >200 clones, likely because RAW264.7 cells are not easy to transfect².

3. In Supplementary Fig. 1 and corresponding Methods section, please change the average expression value to individual expression values, since this is a very small cohort (are there any larger cohorts that can be included in this study?) and any outlier value would potentially affect the average value.

Re: We thank this reviewer for this suggestion. Accordingly, we downloaded the source data from the database instead of using the associated online tool. Analysis of the source data showed that in both humans and mice, MPPs had significantly lower expression of MALAT1 than HSCs (**new data, Supplementary Fig. 1b, d**).

4. In Fig. 1e-g, in addition to RAW264.7, have the authors tried to use human macrophage/monocyte cell line to examine the role of Malat1 during osteoclastogenesis?

Re: We thank this reviewer for this suggestion. To extend our investigation to human osteoclastogenesis, we treated the U937 human pre-osteoclast/monocyte cell line with PMA (100 ng/mL) for 2 days, followed by 12-14 days of human M-CSF (50 ng/mL) and RANKL (100 ng/mL) treatment, as described previously^{3,4}. NFATC1 expression showed an initial upregulation in the first 5 days, followed by a decrease, while the osteoclast marker TRAP exhibited a progressive elevation (**new data, Supplementary Fig. 7a**). To determine the role of MALAT1 in human osteoclast differentiation, we used CRISPRi to knock down MALAT1. Five sgRNAs (sg1-5) that target the human *MALAT1* promoter were tested in HEK293T cells, and four out of five sgRNAs showed approximately 50% knockdown efficiency (**new data, Supplementary Fig. 7b**). We used sg2 and sg4 to deplete MALAT1 in U937 cells, achieving over 95% knockdown efficiency in this cell line (**new data, Supplementary Fig. 7c**). Subsequently, osteoclast differentiation assays revealed a higher number of TRAP-positive osteoclasts in the MALAT1 knockdown group compared with the control group (**new data, Supplementary Fig. 7d, e**). Moreover, NFATC1 and TRAP expression levels were elevated in MALAT1-knockdown U937 cells during osteoclast differentiation (**new data, Supplementary Fig. 7f-h**). Hence, MALAT1 functions as a suppressor of both mouse and human osteoclastogenesis.

Similar to RAW264.7 cells (Fig. 6e), during osteoclast differentiation of U937 cells, TEAD3 was also the most upregulated TEAD family member (**new data, Fig. 6f**). Thus, we also examined TEAD3's function in U937 cells, finding that shRNA-mediated knockdown of TEAD3 impaired human osteoclastogenesis (**new data, Supplementary Fig. 10a-c**) and downregulated the expression of NFATC1 and TRAP at both mRNA and protein levels (**new data, Supplementary Fig. 10d-f**). In addition, MALAT1 depletion did not affect the upregulation of

TEAD3 during osteoclast differentiation (**new data, Supplementary Fig. 10g**), suggesting that MALAT1 does not regulate TEAD3's expression levels (but instead inhibits the TEAD3-NFATC1 interaction).

5. In Fig. 2, have the authors confirmed the expression of Malat1 in different organs of the genetic mouse models?

Re: As requested, we collected various tissues, including bone marrow, stomach, colon, small intestine, liver, and pancreatic tissues, from *Malat1*^{+/+}, *Malat1*^{-/-}, and *Malat1*^{-/-};*Malat1*^{Tg/Tg} mice and measured Malat1 expression levels by qPCR. This analysis confirmed Malat1 depletion in *Malat1*^{-/-} mice and its re-expression in *Malat1*^{-/-};*Malat1*^{Tg/Tg} mice, although the levels of Malat1 restoration varied among different tissues (**new data, Supplementary Fig. 2a**).

6. In Fig. 2o, the authors used one B16F1 melanoma cell intratibial bone metastasis model. Intratibial injection is not an ideal method to study bone metastasis, even though I understand the other methods such as intracardiac or intra-caudal arterial may present some technical issues. Since bone metastasis is one of the major conclusions, this should be more carefully addressed. Maybe the authors can try different methods or multiple models using the same method. Otherwise, I'm not confident this data is sufficient to support the conclusion that Malat1 deficiency promotes bone metastasis.

Re: We performed intratibial injections because this study focused on the role of Malat1 in the bone microenvironment (or more specifically, in the osteoclast lineage). Since Malat1 knockout mice developed osteoporosis and showed an increase in osteoclasts (Figure 2), we focused on testing the hypothesis that the precancer osteoporosis and increased osteoclasts in *Malat1*^{-/-} mice would promote bone metastasis after tumor cells seed the bone marrow. This hypothesis is clinically relevant.

We fully agree with this reviewer that it is important to corroborate our findings in multiple metastasis models. To this end, we performed intratibial injection of 3-month-old female *Malat1*^{+/+}, *Malat1*^{-/-}, and *Malat1*^{-/-};*Malat1*^{Tg/Tg} mice with the EO771 cell line, a cell line derived from a mouse mammary tumor on a C57BL/6 background⁶. Before injecting tumor cells, we conducted μ CT scanning and confirmed that at this age, only female *Malat1*^{-/-} mice, but not female *Malat1*^{+/+} and *Malat1*^{-/-};*Malat1*^{Tg/Tg} mice, exhibited signs of osteoporosis (**new data, Supplementary Fig. 2j-n**).

After injection with 2×10^5 luciferase-labeled EO771 cells, bioluminescent signals showed no significant difference in baseline levels among the three animal groups on the injection day. At the endpoint, we observed significantly higher signals in *Malat1*^{-/-} mice compared with *Malat1*^{+/+} and *Malat1*^{-/-};*Malat1*^{Tg/Tg} mice (**new data, Fig. 3e and Supplementary Fig. 2o**). After euthanasia, we collected the tibiae for *ex vivo* imaging (**new data, Fig. 3f, g**), which confirmed *in vivo* imaging results. We also performed X-ray imaging of the tibiae and found that *Malat1*^{-/-} mice had more osteolytic lesions (**new data, Fig. 3h**). Moreover, H&E staining of bone sections demonstrated higher tumor burdens in the tibiae of *Malat1*^{-/-} mice, as evidenced by more cancerous lesions in the cortical bone near the growth plate and deeper extension of tumor areas into the distal bone marrow cavity (**new data, Fig. 3i**). Immunohistochemical staining of RFP (co-expressed with luciferase) supported the histologic analysis (**new data, Fig. 3i**). In addition, TRAP staining revealed elevated osteoclast numbers in the tibiae of *Malat1*^{-/-} mice compared with *Malat1*^{+/+} and *Malat1*^{-/-};*Malat1*^{Tg/Tg} mice (**new data, Fig. 3j-l**). Taken together with the results from the B16F1 model (Fig. 3a-d), these findings collectively suggest that loss of Malat1 in host mice exacerbates metastatic bone colonization by melanoma and mammary tumor cells.

7. The genetic model authors used are appropriate to test their hypothesis. However, the rescue phenotypes are not exactly same in male mice and female mice (Extended Fig. 2a, b, c, d). The authors claim that Malat1 has functional relationship with AR in prostate cancer and generally women are more at risk of developing osteoporosis than men. Have the authors examined a role for AR along with Malat1 in osteoporosis?

Re: As this reviewer correctly pointed out, female *Malat1*^{-/-} mice exhibited a more severe osteoporotic phenotype and less phenotype rescue by Malat1 re-expression. It is also true that in general, women are more at risk of developing osteoporosis than men. Thus, it will be of interest to examine whether sex hormones and their receptors (ER, PR, and AR) regulate bone density through Malat1. This is outside the scope of the present manuscript and will be a topic of our follow-up study. Therefore, in the revised manuscript, we do not make any claim about the potential relationship between Malat1 and sex hormone receptors like AR.

8. Fig. 3g shows outliers in two knockdown groups. If the outliers were excluded, the average would not be significantly different than controls.

Re: We removed the outliers in the two knockdown groups, and the differences between the knockdown and the control are still statistically significant (**revised data, Fig. 5g**).

9. The suggestion that the nuclear accumulation of Nfatc1 in Extended Data Fig. 4d is not convincing since the difference is subtle and perhaps not run on the same gel? Total lysate bands should be compared on the same gel.

Re: We agree with this reviewer. Accordingly, we repeated this fractionation experiment and ran total lysates, nuclear fractions, and cytoplasmic fractions on the same gel. We found that knockdown of Malat1 in RAW264.7 cells led to an increase in Nfatc1 protein levels in all three fractions after RANKL treatment (**new data, Supplementary Fig. 5d**). This is consistent with the fact that Nfatc1 activates its own expression.

10. In Fig. 4b, it seems that Tead3 may be selectively expressed in BMMs, but RAW264.7 displays Tead-2, 3, 4 expression. And L929 seems to have a higher Tead3 expression. Please correct corresponding statements in the text.

Re: Since RAW264.7 is a cell line that has been passaged *in vitro*, it is not surprising that it has a different expression pattern (i.e., displaying Tead-2, 3, 4 expression) from primary BMMs. Thus, we have revised the statement to “Tead3 showed a relatively specific expression pattern in primary BMMs (Fig. 6a)”.

11. There are several other proteins that also interact with Nfatc1 (Extended data Fig. 5a). The authors should test other candidates as well as Tead3 in their experiments to show that the effects are specific to Nfatc1 and Tead3 interaction.

Re: In this panel (now Supplementary Fig. 8a), besides TEAD, we also circled the AP1 complex components (JUN and FOS) and CREB1, because they have been reported to regulate osteoclast differentiation^{23, 24}. As requested by this reviewer, we pulled down NFATC1 from the control, MALAT1-knockout, and Malat1-restored HEK293T cells, followed by immunoblotting with antibodies against FOS, JUN, and CREB1. While we did not detect an interaction of FOS with NFATC1, we observed interactions of JUN and CREB1 with NFATC1; however, unlike the TEAD3-NFATC1 interaction, these interactions were not affected by MALAT1 (**new data, Supplementary Fig. 9d, e**). This result further justifies our focus on TEAD3, whose interaction with NFATC1 was reduced by MALAT1 (Fig. 6h-k).

12. The ChIP-qPCR verified the occupancy at the Nfatc1 regions. However, the results would be more significant if they carry out ChIP-seq at the genome-wide level instead of a single target.

Re: We agree that instead of showing a single Nfatc1 target gene, it would be better to examine other Nfatc1 target genes. However, doing Nfatc1 ChIP-seq with RAW264.7 cells turned out to be challenging. In this field, the Nfatc1 targets involved in osteoclast differentiation have been well studied; moreover, the binding sites in the promoters have been identified for several target genes, including *Ctsk*, *Acp5*, and *Nfatc1* itself²⁵⁻²⁷. Therefore, in addition to the *Ctsk* data, we also performed ChIP-qPCR to examine Nfatc1 occupancy of *Acp5* and *Nfatc1* gene promoters, finding that after RANKL treatment, Malat1-knockdown RAW264.7 cells showed more occupancy

of *Acp5* and *Nfatc1* gene promoters by *Nfatc1* than the control RAW264.7 cells (**new data, Supplementary Fig. 5e, f**).

13. Based on the hypothesis and model (Fig. 4s), Malat1 acts a decoy suppressor protein. Maybe the authors can try to overexpress Malat1 and treat with RANKL and compare the downstream targets with WT. If overexpression of Malat1 suppresses osteoporosis compared to WT that would further support their hypothesis.

Re: We thank this reviewer for suggesting adding the Malat1 overexpression data. Consistent with Malat1 being a highly abundant lncRNA, we tested various methods and could only overexpress Malat1 in RAW264.7 cells by using the piggyBac transposon system and electroporation. The resulting overexpression level was approximately a 1.7-fold increase over the endogenous expression level (**new data, Supplementary Fig. 6a**), which did not lead to significant differences in RANKL-induced osteoclastogenesis or the expression of *Nfatc1*, *Trap5*, and *Ctsk* (**new data, Supplementary Fig. 6b-g**). Moreover, BMMs from *Malat1*^{Tg/Tg} mice exhibited approximately a 1.5-fold increase in Malat1 expression relative to BMMs from *Malat1*^{+/+} mice (**new data, Supplementary Fig. 6h**). Compared with *Malat1*^{+/+} mice, *Malat1*^{Tg/Tg} mice did not display any significant difference in bone density or other bone parameters based on μ CT analysis (**new data, Supplementary Fig. 6i-m**). The challenge of achieving substantial Malat1 overexpression in wild-type cells and mice limited a comprehensive examination of its overexpression effects. However, considering that reduced MALAT1 expression in pre-osteoclasts and osteoclasts is associated with osteoporosis and bone metastasis (**new data, Fig. 4i-k**), our loss-of-function approach, coupled with re-expression of Malat1 in Malat1-deficient mice and cell lines, is suitable for this investigation.

14. Is there any clinical data that would support the role of Malat1 in osteoporosis? Do patients who have osteoporosis display low expression levels of Malat1?

Re: We fully agree with this reviewer that it is important to assess the clinical relevance of MALAT1 in osteoporosis and bone metastasis. To this end, we analyzed single-cell RNA-seq data from human bone tissues. The datasets included GSE190772 with samples from two patients with breast cancer bone metastasis^{5, 6}, GSE162454 with samples from six osteosarcoma patients^{7, 8}, and GSE169396 featuring bone tissues from a non-osteoporotic individual and three osteoporosis patients (femoral head collected during hip replacement surgery)⁹. Osteosarcomas and breast cancer bone metastases often exhibit osteolytic features.

We used the “Harmony” method¹⁰ to remove batch effects between samples, subsequently applying dimensionality reduction to annotate cell types based on marker genes (**new data, Supplementary Fig. 4a-c**). These analyses defined the cell cluster-specific transcriptome of different patient groups. We then analyzed the expression of MALAT1 in pre-osteoclasts (including monocytes and macrophages) and mature osteoclasts of the non-osteoporotic individual (**new data, Fig. 4a, b**), osteoporosis patients (**new data, Fig. 4c, d**), osteosarcoma patients (**new data, Fig. 4e, f**), and patients with breast cancer bone metastases (**new data, Fig. 4g, h**). Within each group, MALAT1 expression levels were significantly lower in osteoclasts compared with pre-osteoclasts (**new data, Fig. 4b, d, f, and h**). Moreover, across the four patient groups, MALAT1 expression levels in pre-osteoclasts and osteoclasts were significantly lower in patients with osteoporosis, osteosarcoma, or breast cancer bone metastasis than in the non-osteoporotic individual (**new data, Fig. 4i-k**). These findings indicate that reduced MALAT1 expression in the osteoclast lineage is associated with osteoporosis and bone lesions, including breast cancer metastases and osteosarcomas.

Minor:

1. Some main Figures contain too many sub-Figures and appear too crowded. The authors should consider moving some of them to extended Figures.

Re: We have reorganized the figures.

References

1. Kim J, Piao HL, Kim BJ, Yao F, Han Z, Wang Y, Xiao Z, Siverly AN, Lawhon SE, Ton BN, Lee H, Zhou Z, Gan B, Nakagawa S, Ellis MJ, Liang H, Hung MC, You MJ, Sun Y, Ma L. Long noncoding RNA MALAT1 suppresses breast cancer metastasis. *Nat Genet.* 2018;50(12):1705-15. PMID: PMC6265076.
2. Carralot JP, Kim TK, Lenseigne B, Boese AS, Sommer P, Genovesio A, Brodin P. Automated high-throughput siRNA transfection in raw 264.7 macrophages: a case study for optimization procedure. *J Biomol Screen.* 2009;14(2):151-60.
3. Kang K, Nam S, Kim B, Lim JH, Yang Y, Lee MS, Lim JS. Inhibition of osteoclast differentiation by overexpression of NDRG2 in monocytes. *Biochem Biophys Res Commun.* 2015;468(4):611-6.
4. Ghosh M, Kelava T, Madunic IV, Kalajzic I, Shapiro LH. CD13 is a critical regulator of cell-cell fusion in osteoclastogenesis. *Sci Rep.* 2021;11(1):10736. PMID: PMC8144195.
5. Ding K, Chen F, Priedigkeit N, Brown DD, Weiss K, Watters R, Levine KM, Heim T, Li W, Hooda J, Lucas PC, Atkinson JM, Oesterreich S, Lee AV. Single cell heterogeneity and evolution of breast cancer bone metastasis and organoids reveals therapeutic targets for precision medicine. *Ann Oncol.* 2022;33(10):1085-8. PMID: PMC10007959.
6. Faget DV, Luo X, Inkman MJ, Ren Q, Su X, Ding K, Waters MR, Raut GK, Pandey G, Dodhiawala PB, Ramalho-Oliveira R, Ye J, Cole T, Murali B, Zheleznyak A, Shokeen M, Weiss KR, Monahan JB, DeSelm CJ, Lee AV, Oesterreich S, Weilbaecher KN, Zhang J, DeNardo DG, Stewart SA. p38MAPK α Stromal Reprogramming Sensitizes Metastatic Breast Cancer to Immunotherapy. *Cancer Discov.* 2023;13(6):1454-77. PMID: PMC10238649.
7. Liu Y, Feng W, Dai Y, Bao M, Yuan Z, He M, Qin Z, Liao S, He J, Huang Q, Yu Z, Zeng Y, Guo B, Huang R, Yang R, Jiang Y, Liao J, Xiao Z, Zhan X, Lin C, Xu J, Ye Y, Ma J, Wei Q, Mo Z. Single-Cell Transcriptomics Reveals the Complexity of the Tumor Microenvironment of Treatment-Naive Osteosarcoma. *Front Oncol.* 2021;11:709210. PMID: PMC8335545.
8. Rothzerg E, Feng W, Song D, Li H, Wei Q, Fox A, Wood D, Xu J, Liu Y. Single-Cell Transcriptome Analysis Reveals Paraspeckles Expression in Osteosarcoma Tissues. *Cancer Inform.* 2022;21:11769351221140101. PMID: PMC9730017.
9. Qiu X, Liu Y, Shen H, Wang Z, Gong Y, Yang J, Li X, Zhang H, Chen Y, Zhou C, Lv W, Cheng L, Hu Y, Li B, Shen W, Zhu X, Tan LJ, Xiao HM, Deng HW. Single-cell RNA sequencing of human femoral head in vivo. *Aging (Albany NY).* 2021;13(11):15595-619. PMID: PMC8221309.
10. Korsunsky I, Millard N, Fan J, Slowikowski K, Zhang F, Wei K, Baglaenko Y, Brenner M, Loh PR, Raychaudhuri S. Fast, sensitive and accurate integration of single-cell data with Harmony. *Nat Methods.* 2019;16(12):1289-96. PMID: PMC6884693.
11. Ivanov IP, Firth AE, Michel AM, Atkins JF, Baranov PV. Identification of evolutionarily conserved non-AUG-initiated N-terminal extensions in human coding sequences. *Nucleic Acids Res.* 2011;39(10):4220-34. PMID: PMC3105428.
12. Diaz de Arce AJ, Noderer WL, Wang CL. Complete motif analysis of sequence requirements for translation initiation at non-AUG start codons. *Nucleic Acids Res.* 2018;46(2):985-94. PMID: PMC5778536.
13. Al-Barghouthi BM, Mesner LD, Calabrese GM, Brooks D, Tommasini SM, Bouxsein ML, Horowitz MC, Rosen CJ, Nguyen K, Haddox S, Farber EA, Onengut-Gumuscu S, Pomp D, Farber CR. Systems genetics in diversity outbred mice inform BMD GWAS and identify determinants of bone strength. *Nat Commun.* 2021;12(1):3408. PMID: PMC8184749.
14. Morris JA, Kemp JP, Youlten SE, Laurent L, Logan JG, Chai RC, Vulpescu NA, Forgetta V, Kleinman A, Mohanty ST, Sergio CM, Quinn J, Nguyen-Yamamoto L, Luco AL, Vijay J, Simon MM, Pramatarova A, Medina-Gomez C, Trajanoska K, Ghirardello EJ, Butterfield NC, Curry KF, Leitch VD, Sparkes PC, Adoum AT, Mannan NS, Komla-Ebri DSK, Pollard AS, Dewhurst HF, Hassall TAD, Beltejar MG, Adams DJ, Vaillancourt SM, Kaptoge S, Baldock P, Cooper C, Reeve J, Ntzani EE, Evangelou E, Ohlsson C, Karasik D, Rivadeneira F, Kiel DP, Tobias JH, Gregson CL, Harvey NC, Grundberg E, Goltzman D, Adams DJ, Lelliott CJ, Hinds DA, Ackert-Bicknell CL, Hsu YH, Maurano MT, Croucher PI, Williams

- GR, Bassett JHD, Evans DM, Richards JB. An atlas of genetic influences on osteoporosis in humans and mice. *Nat Genet.* 2019;51(2):258-66. PMID: PMC6358485.
15. Younes N, Syed N, Yadav SK, Haris M, Abdallah AM, Abu-Madi M. A Whole-Genome Sequencing Association Study of Low Bone Mineral Density Identifies New Susceptibility Loci in the Phase I Qatar Biobank Cohort. *J Pers Med.* 2021;11(1). PMID: PMC7825795.
 16. Brooks PJ, Glogauer M, McCulloch CA. An Overview of the Derivation and Function of Multinucleated Giant Cells and Their Role in Pathologic Processes. *Am J Pathol.* 2019;189(6):1145-58.
 17. Xie H, Liao X, Chen Z, Fang Y, He A, Zhong Y, Gao Q, Xiao H, Li J, Huang W, Liu Y. LncRNA MALAT1 Inhibits Apoptosis and Promotes Invasion by Antagonizing miR-125b in Bladder Cancer Cells. *J Cancer.* 2017;8(18):3803-11. PMID: PMC5688934.
 18. Stojic L, Lun ATL, Mangei J, Mascalchi P, Quarantotti V, Barr AR, Bakal C, Marioni JC, Gergely F, Odom DT. Specificity of RNAi, LNA and CRISPRi as loss-of-function methods in transcriptional analysis. *Nucleic Acids Res.* 2018;46(12):5950-66. PMID: PMC6093183.
 19. Much C, Auchynnikava T, Pavlinic D, Buness A, Rappsilber J, Benes V, Allshire R, O'Carroll D. Endogenous Mouse Dicer Is an Exclusively Cytoplasmic Protein. *PLoS Genet.* 2016;12(6):e1006095. PMID: PMC4890738.
 20. Hagedorn PH, Pontoppidan M, Bisgaard TS, Berrera M, Dieckmann A, Ebeling M, Moller MR, Hudlebusch H, Jensen ML, Hansen HF, Koch T, Lindow M. Identifying and avoiding off-target effects of RNase H-dependent antisense oligonucleotides in mice. *Nucleic Acids Res.* 2018;46(11):5366-80. PMID: PMC6009603.
 21. Lai F, Damle SS, Ling KK, Rigo F. Directed RNase H Cleavage of Nascent Transcripts Causes Transcription Termination. *Mol Cell.* 2020;77(5):1032-43 e4.
 22. Lee JS, Mendell JT. Antisense-Mediated Transcript Knockdown Triggers Premature Transcription Termination. *Mol Cell.* 2020;77(5):1044-54 e3. PMID: PMC7093083.
 23. Takayanagi H. The role of NFAT in osteoclast formation. *Ann N Y Acad Sci.* 2007;1116:227-37.
 24. Kim JH, Kim N. Regulation of NFATc1 in Osteoclast Differentiation. *J Bone Metab.* 2014;21(4):233-41. PMID: PMC4255043.
 25. Balkan W, Martinez AF, Fernandez I, Rodriguez MA, Pang M, Troen BR. Identification of NFAT binding sites that mediate stimulation of cathepsin K promoter activity by RANK ligand. *Gene.* 2009;446(2):90-8.
 26. Fretz JA, Shevde NK, Singh S, Darnay BG, Pike JW. Receptor activator of nuclear factor-kappaB ligand-induced nuclear factor of activated T cells (C1) autoregulates its own expression in osteoclasts and mediates the up-regulation of tartrate-resistant acid phosphatase. *Mol Endocrinol.* 2008;22(3):737-50. PMID: PMC2262172.
 27. Li X, Islam S, Xiong M, Nsumu NN, Lee MW, Jr., Zhang LQ, Ueki Y, Heruth DP, Lei G, Ye SQ. Epigenetic regulation of NfatC1 transcription and osteoclastogenesis by nicotinamide phosphoribosyl transferase in the pathogenesis of arthritis. *Cell Death Discov.* 2019;5:62. PMID: PMC6365567.

REVIEWERS' COMMENTS

Reviewer #2 (Remarks to the Author):

The authors have satisfactorily addressed my initial comments.

Reviewer #3 (Remarks to the Author):

In the revised version of the manuscript, the authors have updated the date and date description with reasonably addressing comments and concerns raised by the reviewers.

Reviewer #4 (Remarks to the Author):

In this revised manuscript submitted to Nature Communications, the authors have responded to all questions raised by original submission, including adding more in vitro and in vivo model experiments to better support their original findings. Below are suggested minor revisions of the current manuscript prior to publication.

Minor revision:

1. Overall, in order to obtain unbiased, comprehensive view of protein interactions, mass spectrometry would be the better approach rather than testing individual protein-protein interaction.
2. The authors predominantly focused on Malat1 due to prior research in their group, emphasizing that "the lncRNA MALAT1 sequesters TEAD proteins to suppress breast cancer lung metastasis." However, this doesn't inherently support the potential role of MALAT1 in osteoporosis. To strengthen their hypothesis, I suggest a restructure of the Introduction section to provide a more extensive background on MALAT1's involvement in decreased bone mineral density, positioning this earlier in the manuscript.
3. Regarding the new Supplementary Fig. 5d, could the authors potentially modify the RANKL treatment time or concentration to accentuate the difference between control and KD? Alternatively, quantifying the blots might enhance clarity.
4. In response to the rebuttal of the new Fig. 6a, it's unclear whether the authors are suggesting an association between the expression pattern of Tead proteins and in vitro culturing conditions. It seems the focus of this experiment is Tead-3, given its relatively higher expression compared to other Tead proteins. I recommend rephrasing the statements to clarify this aspect.

Point-by-point rebuttal to the reviewers

Nature Communications manuscript NCOMMS-23-02783A (Accepted in Principle)

Reviewer #2:

The authors have satisfactorily addressed my initial comments.

Re: We thank this reviewer for finding our revision satisfactory.

Reviewer #3:

In the revised version of the manuscript, the authors have updated the data and data description with reasonably addressing comments and concerns raised by the reviewers.

Re: We thank this reviewer for appreciating that we have addressed the previous concerns.

Reviewer #4:

1. Overall, in order to obtain unbiased, comprehensive view of protein interactions, mass spectrometry would be the better approach rather than testing individual protein-protein interaction.

Re: We agree that mass spectrometry provides an unbiased view of protein interactions. It is worth mentioning that NFATC1 is a well-studied protein, and that the list of NFATC1-interacting proteins obtained from the protein-protein interaction database, Mentha, was based on mass spectrometric results.

2. The authors predominantly focused on Malat1 due to prior research in their group, emphasizing that "the lncRNA MALAT1 sequesters TEAD proteins to suppress breast cancer lung metastasis." However, this doesn't inherently support the potential role of MALAT1 in osteoporosis. To strengthen their hypothesis, I suggest a restructure of the Introduction section to provide a more extensive background on MALAT1's involvement in decreased bone mineral density, positioning this earlier in the manuscript.

Re: We agree with this reviewer that our previous Nature Genetics paper, which reported that MALAT1 suppresses breast cancer metastasis, does not inherently support the potential role of MALAT1 in osteoporosis. However, in our opinion, the structure and logical flow of the Introduction section are suitable for this paper, because the first two paragraphs provide a brief introduction to osteoporosis and the next two paragraphs provide a background on MALAT1 and its potential involvement in decreased bone mineral density. This not only offers the rationale for studying the role of MALAT1 in osteoporosis and bone metastasis, but also allows a smooth transition to the Results section.

3. Regarding the new Supplementary Fig. 5d, could the authors potentially modify the RANKL treatment time or concentration to accentuate the difference between control and KD? Alternatively, quantifying the blots might enhance clarity.

Re: As requested, we quantified the signal intensity in Supplementary Fig. 5d by using ImageJ.

4. In response to the rebuttal of the new Fig. 6a, it's unclear whether the authors are suggesting an association between the expression pattern of Tead proteins and in vitro culturing conditions. It seems the focus of this experiment is Tead-3, given its relatively higher expression compared to other Tead proteins. I recommend rephrasing the statements to clarify this aspect.

Re: We focused on Tead3 instead of other Tead family members not only because "Tead3 showed a relatively specific expression pattern in primary BMMs (Fig. 6a)", but also because "RANKL treatment of RAW264.7 and U937 cells upregulated Tead3, but not other Tead family members (Fig. 6e, f)". Both quoted statements are copied verbatim from the Results section of our manuscript. Thus, the rationale for focusing on Tead3 is crystal clear.